# Towards Model-Agnostic Federated Learning Using Knowledge Distillation

**Andrei Afonin**
EPFL
andrei.afonin@epfl.ch

**Sai Praneeth Karimireddy**[*]
EPFL, UC Berkeley
sp.karimireddy@berkeley.edu

## Abstract

Is it possible to design an *universal API* for federated learning using which an ad-hoc group of data-holders (agents) collaborate with each other and perform federated learning? Such an API would necessarily need to be *model-agnostic* i.e. make no assumption about the model architecture being used by the agents, and also cannot rely on having representative public data at hand. *Knowledge distillation* (KD) is the obvious tool of choice to design such protocols. However, surprisingly, we show that most natural KD-based federated learning protocols have poor performance.

To investigate this, we propose a new theoretical framework, Federated Kernel ridge regression, which can capture both model heterogeneity as well as data heterogeneity. Our analysis shows that the degradation is largely due to a fundamental limitation of knowledge distillation under *data heterogeneity*. We further validate our framework by analyzing and designing new protocols based on KD. Their performance on real world experiments using neural networks, though still unsatisfactory, closely matches our theoretical predictions.

## 1 Introduction

> "I speak and speak, but the listener retains only the words he is expecting... It is not the voice that commands the story: it is the ear." - Invisible Cities, Italo Calvino.

Federated learning (and more generally collaborative learning) involves multiple data holders (whom we call agents) collaborating with each other to train their machine learning model over their collective data. Crucially, this is done without directly exchanging any of their raw data (McMahan et al., 2017; Kairouz et al., 2019). Thus, communication is limited to only what is essential for the training process and the data holders (aka agents) retain full ownership over their datasets.

Algorithms for this setting such as FedAvg or its variants all proceed in rounds (Wang et al., 2021). In each such round, the agents first train their models on their local data. Then, the knowledge from these different models is aggregated by *averaging the parameters*. However, exchanging knowledge via averaging the model parameters is only viable if all the agents use the same model architecture. This fundamental assumption is highly restrictive. Different agents may have different computational resources and hence may want to use different model architectures. Further, directly averaging the model parameters can fail even when all clients have the same architecture (Wang et al., 2019b; Singh & Jaggi, 2020; Yu et al., 2021). This is because the loss landscape of neural networks is highly non-convex and has numerous symmetries with different parameter values representing the same function. Finally, these methods are also not applicable when using models which are not based on gradient descent such as random forests. To overcome such limitations, we would need to take a *functional view* view of neural networks i.e. we need methods that are agnostic to the model architecture and parameters. This motivates the central question investigated in this work:

Can we design *model agnostic* federated learning algorithms which would allow each agent to train their model of choice on the combined dataset?

---

[*]Corresponding author.

Specifically, we restrict the algorithms to access the models using only two primitives (a universal model API): train on some dataset i.e. *fit*, and yield predictions on some inputs i.e. *predict*. Our goal is to be able to collaborate with and learn from any agent which provides these two functionalities.

**Simple algorithms.** A naive such model agnostic algorithm indeed exists—agents can simply transfer their entire training data to each other and then each agent can train any model of choice on the combined dataset. However, transferring of the dataset is disallowed in federated learning. Instead, we will replace the averaging primitive in federated learning with *knowledge distillation* (KD) (Bucilua et al., 2006; Hinton et al., 2015). In knowledge distillation (KD), information is transferred from model A to model B by training model B on the predictions of model A on some data. Since we only access model A through its predictions, KD is a functional model-agnostic protocol. The key challenge of KD however is that it is poorly understood and cannot be formulated in the standard stochastic optimization framework like established techniques (Wang et al., 2021). Thus, designing and analyzing algorithms that utilize KD requires developing an entirely new framework and approach.

**Our Contributions.** The main results in this work are

- We formulate the model agnostic learning problem as two agents with local datasets wanting to perform kernel regression on their combined datasets. Kernel regression is both simple enough to be theoretically tractable and rich enough to capture non-linear function fitting thereby allowing each agent to have a different kernel (hence different models).
- We analyze alternating knowledge distillation (AKD) and show that it is closely linked to the alternating projection method for finding the intersection of convex sets. Our analysis reveals that AKD sequentially loses information, leading to degradation of performance. This degradation is especially severe when the two agents have heterogeneous data.
- Using the connection to alternating projection, we analyze other possible variants such as *averaged* knowledge distillation (AvgKD) and attempt to construct an 'optimal' scheme.
- Finally, we evaluate all algorithms on real world deep learning models and datasets, and show that the empirical behavior closely matches our insights from the theoretical analysis. This demonstrates the utility of our framework for analyzing and designing new algorithms.

## 2 RELATED WORK

**Federated learning (FL).** In FL (Kairouz et al., 2019), training data is distributed over several agents or locations. For instance, these could be several hospitals collaborating on a clinical trial, or billions of mobile phones involved in training a voice recognition application. The purpose of FL is to enable training on the union of all agents' individual data without needing to transmit any of the raw sensitive data. Typically, the training is coordinated by some trusted server. One can also instead use direct peer-to-peer communications (Nedic, 2020). A large body of work has designed algorithms for FL under the identical model setting where we either learn a single global model (McMahan et al., 2017; Reddi et al., 2020; Karimireddy et al., 2020b;a; Wang et al., 2021), or multiple personalized models (Wang et al., 2019a; Deng et al., 2020; Mansour et al., 2020; Grimberg et al., 2021).

**Knowledge distillation (KD).** Initially, KD was introduced as a way to compress models i.e. as a way to transfer the knowledge of a large model to a smaller model (Bucilua et al., 2006; Hinton et al., 2015). Since then, it has found much broader applications such as improving generalization performance via self-distillation, learning with noisy data, and transfer learning (Yim et al., 2017). We refer to a recent survey (Gou et al., 2021) for progress in this vast area.

**KD in FL.** Numerous works propose to use KD to transfer knowledge from the agent models to a centralized server model (Seo et al., 2020; Sattler et al., 2020; Lin et al., 2020; Li et al., 2020; Wu et al., 2021). However, all of these methods rely on access to some common public dataset which may be impractical. KD has also been proposed to combine personalization with model compression (Ozkara et al., 2021), but our focus is for the agents to learn on the combined data. In the closely related *codistillation* setting (Zhang et al., 2018; Anil et al., 2018; Sodhani et al., 2020), an ensemble of students learns collaboratively without a central server model. While codistillation does not need additional unlabelled data, it is only suitable for distributed training within a datacenter since it assumes all agents have access to the same dataset. In FL however, there is both model and data heterogeneity. Further, none of these methods have a theoretical analysis.

**KD analysis.** Despite the empirical success of KD, it is poorly understood with very little theoretical analysis. Phuong & Lampert (2021) explore a generalization bound for a distillation-trained linear model, and Tang et al. (2021) conclude that by using KD one re-weights the training examples for the student, and Menon et al. (2020) consider a Bayesian view showing that the student learns better if the teacher provides the true Bayes probability distribution. Allen-Zhu & Li (2020) show how ensemble distillation can preserve their diversity in the student model. Finally, Mobahi et al. (2020) consider self-distillation in a kernel regression setting i.e. the model is retrained using its own predictions on the training data. They show that iterative self-distillation induces a strong regularization effect. We significantly extend their theoretical framework in our work to analyze KD in federated learning where agents have different models and different datasets.

## 3 FRAMEWORK AND SETUP

**Notation.** We denote a set as $\mathcal{A}$, a matrix as $\boldsymbol{A}$, and a vector as $\boldsymbol{a}$. $\boldsymbol{A}[i,j]$ is the (i, j)-th element of matrix $\boldsymbol{A}$, $\boldsymbol{a}[i]$ denotes the i'th element of vector $\boldsymbol{a}$. $||\boldsymbol{a}||$ denotes the $\ell_2$ norm of vector $\boldsymbol{a}$.

**Centralized kernel regression (warmup).** Consider, as a warm-up, the centralized setting with a training dataset $\mathcal{D} \subseteq \mathbb{R}^d \times \mathbb{R}$. That is, $\mathcal{D} = \cup_i^N \{(\boldsymbol{x}_i, y_i)\}$, where $\boldsymbol{x}_n \in \mathcal{X} \subseteq \mathbb{R}^d$ and $y_n \in \mathcal{Y} \subseteq \mathbb{R}$. Given training set $\mathcal{D}$, our aim is to find best function $f^\star \in \mathcal{F} : \mathcal{X} \to \mathcal{Y}$. To find $f^\star$ we solve the following regularized optimization problem:

$$f^\star := \arg\min_{f \in \mathcal{F}} \frac{1}{N} \sum_n (f(\boldsymbol{x}_n) - y_n)^2 + cR_u(f), \text{ with} \tag{1}$$

$$R_u(f) := \int_{\mathcal{X}} \int_{\mathcal{X}} u(\boldsymbol{x}, \boldsymbol{x}')f(\boldsymbol{x})f(\boldsymbol{x}')d\boldsymbol{x}d\boldsymbol{x}'. \tag{2}$$

Here, $\mathcal{F}$ is defined to be the space of all functions such that (2) is finite, $c$ is the regularization parameter, and $u(\boldsymbol{x}, \boldsymbol{x}')$ is a kernel function. That is, $u$ is symmetric $u(\boldsymbol{x}, \boldsymbol{x}') = u(\boldsymbol{x}', \boldsymbol{x})$ and positive with $R_u(f) = 0$ only when $f = 0$ and $R_u(f) > 0$ o.w. Further, let $k(\boldsymbol{x}, \boldsymbol{t})$ be the function s.t.

$$\int_{\mathcal{X}} u(\boldsymbol{x}, \boldsymbol{x}')k(\boldsymbol{x}', \boldsymbol{t})d\boldsymbol{x}' = \delta(\boldsymbol{x} - \boldsymbol{t}), \quad \text{where } \delta(\cdot) \text{ is the Dirac delta function.} \tag{3}$$

Now, we can define the positive definite matrix $\boldsymbol{K} \in \mathbb{R}^{N \times N}$ and vector $\boldsymbol{k}_{\boldsymbol{x}} \in \mathbb{R}^N$ as:

$$\boldsymbol{K}[i,j] := \frac{1}{N}k(\boldsymbol{x}_i, \boldsymbol{x}_j) \quad \text{and} \quad \boldsymbol{k}_{\boldsymbol{x}}[i] := \frac{1}{N}k(\boldsymbol{x}, \boldsymbol{x}_i), \quad \text{for} \quad \boldsymbol{x}_i \in \mathcal{D}, \forall i \in [N]. \tag{4}$$

Note that $\boldsymbol{k}_{\boldsymbol{x}}$ is actually a vector valued function which takes any $\boldsymbol{x} \in \mathcal{X}$ as input, and both $\boldsymbol{k}_{\boldsymbol{x}}$ and $\boldsymbol{K}$ depend on the training data $\mathcal{D}$. We can then derive a closed form solution for $f^\star$.

**Proposition I** (Schölkopf et al. (2001)). *The $f^\star$ which minimizes (1) is given by*

$$f^\star(\boldsymbol{x}) = \boldsymbol{k}_{\boldsymbol{x}}^\top(c\boldsymbol{I} + \boldsymbol{K})^{-1}\boldsymbol{y}, \quad for \quad \boldsymbol{y}[i] := y_i, \forall i \in [N].$$

Note that on the training data $\boldsymbol{X} \in \mathbb{R}^{N \times d}$ with $\boldsymbol{X}[i, :] = \boldsymbol{x}_i$, we have $f^\star(\boldsymbol{X}) = \boldsymbol{K}(c\boldsymbol{I} + \boldsymbol{K})^{-1}\boldsymbol{y}$. Kernel regression for an input $\boldsymbol{x}$ outputs a weighted average of the training $\{y_n\}$. These weights are computed using a learned measure of distance between the input $\boldsymbol{x}$ and the training $\{\boldsymbol{x}_n\}$. Intuitively, the choice of the kernel $u(\boldsymbol{x}, \boldsymbol{x}')$ creates an inductive bias and corresponds to the choice of a model in deep learning, and the regularization parameter $c$ acts similarly to tricks like early stopping, large learning rate, etc. which help in generalization. When $c = 0$, we completely fit the training data and the predictions exactly recover the labels with $f^\star(\boldsymbol{X}) = \boldsymbol{K}(\boldsymbol{K})^{-1}\boldsymbol{y} = \boldsymbol{y}$. When $c > 0$ the predictions $f^\star(\boldsymbol{X}) = \boldsymbol{K}(c\boldsymbol{I} + \boldsymbol{K})^{-1}\boldsymbol{y} \neq \boldsymbol{y}$ and they incorporate the inductive bias of the model. In knowledge distillation, this extra information carried by the predictions about the inductive bias of the model is popularly referred to as "dark knowledge" (Hinton et al., 2015).

**Federated kernel regression (our setting).** We have two agents, with agent 1 having dataset $\mathcal{D}_1 = \cup_i^{N_1}\{(\boldsymbol{x}_i^1, y_i^1)\}$ and agent 2 with dataset $\mathcal{D}_2 = \cup_i^{N_2}\{(\boldsymbol{x}_i^2, y_i^2)\}$. Agent 1 aims to find the best

approximation mapping $g^{1\star} \in \mathcal{F}_1 : \mathcal{X} \to \mathcal{Y}$ using a kernel $u_1(\boldsymbol{x}, \boldsymbol{x}')$ and objective:

$$g^{1\star} := \arg\min_{g \in \mathcal{F}_1} \frac{1}{N_1 + N_2} \sum_n (g(\boldsymbol{x}_n^1) - y_n^1)^2 + \frac{1}{N_1 + N_2} \sum_n (g(\boldsymbol{x}_n^2) - y_n^2)^2 + cR_{u_1}(g), \text{ with}$$

(5)

$$R_{u_1}(g) := \int_{\mathcal{X}} \int_{\mathcal{X}} u_1(\boldsymbol{x}, \boldsymbol{x}')g(\boldsymbol{x})g(\boldsymbol{x}')d\boldsymbol{x}d\boldsymbol{x}'.$$

(6)

Note that the objective of agent 1 is defined using its *individual kernel* $u_1(\boldsymbol{x}, \boldsymbol{x}')$, but over the *joint dataset* $(\mathcal{D}_1, \mathcal{D}_2)$. Correspondingly, agent 2 also uses its own kernel function $u_2(\boldsymbol{x}, \boldsymbol{x}')$ to define the regularizer $R_{u_2}(g^2)$ over the space of functions $g^2 \in \mathcal{F}_2$ and optimum $g^{2\star}$. Thus, our setting has model heterogeneity (different kernel functions $u_1$ and $u_2$) and data heterogeneity ($\mathcal{D}_1$ and $\mathcal{D}_2$ are not i.i.d.). Given that the setting is symmetric between agents 1 and 2, we can focus solely on error in terms of agent 1's objective (5) without loss of generality.

Proposition I can be used to derive a closed form form for function $g^{1\star}$ minimizing objective (5). However, computing this requires access to the datasets of both agents. Instead, we ask "can we design an iterative federated learning algorithm which can approximate $g^{1\star}$"?

## 4  ALTERNATING KNOWLEDGE DISTILLATION

In this section, we describe a popular iterative knowledge distillation algorithm and analyze its updates in our framework. Our analysis leads to some surprising connections between KD and projection onto convex sets, and shows some limitations of the current algorithm.

**Algorithm.**   Denote the data on agent 1 as $\mathcal{D}_1 = (\boldsymbol{X}^1, \boldsymbol{y}^1)$ where $\boldsymbol{X}^1[i, :] = \boldsymbol{x}_n^1$ and $\boldsymbol{y}^1[i] = y_i^1$. Correspondingly, we have $\mathcal{D}^2 = (\boldsymbol{X}^2, \boldsymbol{y}^2)$. Now starting from $\hat{\boldsymbol{y}}_0^1 = \boldsymbol{y}^1$, in each rounds $t$ and $t + 1$:

   a. Agent 1 trains their model on dataset $(\boldsymbol{X}^1, \hat{\boldsymbol{y}}_t^1)$ to obtain $g_t^1$.
   b. Agent 2 receives $g_t^1$ and uses it to predict labels $\hat{\boldsymbol{y}}_{t+1}^2 = g_t^1(\boldsymbol{X}^2)$.
   c. Agent 2 trains their model on dataset $(\boldsymbol{X}^2, \hat{\boldsymbol{y}}_{t+1}^2)$ to obtain $g_{t+1}^1$.
   d. Agent 1 receives a model $g_{t+1}^1$ from agent 2 and predicts $\hat{\boldsymbol{y}}_{t+2}^1 = g_{t+1}^1(\boldsymbol{X}^1)$.

Thus the algorithm alternates between training and knowledge distillation on each of the two agents. We also summarize the algorithm in Figure 1a. Importantly, note that there is no exchange of raw data but only of the trained models. Further, each agent trains their choice of a model on their data with agent 1 training $\{g_t^1\}$ and agent 2 training $\{g_{t+1}^1\}$. Superscript 1 means we start AKD from agent 1.

### 4.1  THEORETICAL ANALYSIS

Similar to (3), let us define functions $k_1(\boldsymbol{x}, \boldsymbol{x}')$ and $k_2(\boldsymbol{x}, \boldsymbol{x}')$ such that they satisfy

$$\int_{\mathcal{X}} u_a(\boldsymbol{x}, \boldsymbol{x}')k_a(\boldsymbol{x}', \boldsymbol{t})d\boldsymbol{x}' = \delta(\boldsymbol{x} - \boldsymbol{t}) \quad \text{for } a \in \{1, 2\}.$$

For such functions, we can then define the following positive definite matrix $\boldsymbol{L} \in \mathbb{R}^{(N_1+N_2) \times (N_1+N_2)}$:

$$\boldsymbol{L} = \begin{pmatrix} \boldsymbol{L}_{11} & \boldsymbol{L}_{12} \\ \boldsymbol{L}_{21} & \boldsymbol{L}_{22} \end{pmatrix}, \quad \boldsymbol{L}_{a,b}[i, j] = \frac{1}{N_1 + N_2} k_1(\boldsymbol{x}_i^a, \boldsymbol{x}_j^b) \text{ for } a, b \in \{1, 2\} \text{ and } i \in [N_1], j \in [N_2].$$

Note $\boldsymbol{L}$ is symmetric (with $\boldsymbol{L}_{12}^\top = \boldsymbol{L}_{21}$) and is also positive definite. Further, each component $\boldsymbol{L}_{a,b}$ measures pairwise similarities between inputs of agent $a$ and agent $b$ using the kernel $k_1$. Correspondingly, we define $\boldsymbol{M} \in \mathbb{R}^{(N_1+N_2) \times (N_1+N_2)}$ which uses kernel $k_2$:

$$\boldsymbol{M} = \begin{pmatrix} \boldsymbol{M}_{11} & \boldsymbol{M}_{12} \\ \boldsymbol{M}_{21} & \boldsymbol{M}_{22} \end{pmatrix}, \quad \boldsymbol{M}_{a,b}[i, j] = \frac{1}{N_1 + N_2} k_2(\boldsymbol{x}_i^a, \boldsymbol{x}_j^b) \text{ and } i \in [N_1], j \in [N_2].$$

We can now derive the closed form of the AKD algorithm repeatedly using Proposition I.

**Proposition II.** *The model in round 2t learned by the alternating knowledge distillation algorithm is*

$$g_{2t}^1(\boldsymbol{x}) = \boldsymbol{l}_{\boldsymbol{x}}^\top (c\boldsymbol{I} + \boldsymbol{L}_{11})^{-1} \left( \boldsymbol{M}_{12}(c\boldsymbol{I} + \boldsymbol{M}_{22})^{-1} \boldsymbol{L}_{21}(c\boldsymbol{I} + \boldsymbol{L}_{11})^{-1} \right)^t \boldsymbol{y}^1,$$

*where $\boldsymbol{l_x} \in \mathbb{R}^{N_1}$ is defined as $\boldsymbol{l_x}[i] = \frac{1}{N_1+N_2}k_1(\boldsymbol{x}, \boldsymbol{x}_i^1)$. Further, for any fixed $\boldsymbol{x}$ we have*

$$\lim_{t\to\infty} g_t^1(\boldsymbol{x}) = 0\,.$$

First, note that if agents 1 and 2 are identical with the same data and same model, we have $\boldsymbol{M}_{12} = \boldsymbol{M}_{22} = \boldsymbol{L}_{11} = \boldsymbol{L}_{12}$. This setting corresponds to *self-distillation* where the model is repeatedly retrained on its own predictions. Proposition II shows that after $2t$ rounds of self-distillation, we obtain a model is of the form $g_{2t}^1(\boldsymbol{x}) = \boldsymbol{l_x}^\top (c\boldsymbol{I} + \boldsymbol{L}_{11})^{-1} \left(\boldsymbol{L}_{11} (c\boldsymbol{I} + \boldsymbol{L}_{11})^{-1}\right)^{2t} \boldsymbol{y}$. Here, the effect of $c$ is amplified as $t$ increases. Thus, this shows that repeated self-distillation induces a strong regularization effect, recovering the results of (Mobahi et al., 2020).

Perhaps more strikingly, Proposition II shows that not only does AKD fail to converge to the actual optimal solution $g^{1\star}$ as defined in (5), it will also slowly degrade and eventually converges to 0. We next expand upon this and explain this phenomenon.

### 4.2 DEGRADATION AND CONNECTION TO PROJECTIONS

While mathematically, Proposition II completely describes the AKD algorithm, it does not provide much intuition. In this section, we rephrase the result in terms of projections and contractions which provides a more visual understanding of the method.

**Oblique projections.** A projection operator $\boldsymbol{P}$ linearly maps (projects) all inputs onto some linear subspace $\mathcal{A} = \text{Range}(\boldsymbol{A})$. In general, we can always rewrite such a projection operation as

$$\boldsymbol{Px} = \min_{\boldsymbol{y}\in\text{Range}(\boldsymbol{A})} (\boldsymbol{y} - \boldsymbol{x})^\top \boldsymbol{W}(\boldsymbol{y} - \boldsymbol{x}) = \boldsymbol{A}(\boldsymbol{A}^\top \boldsymbol{WA})^{-1}\boldsymbol{A}^\top \boldsymbol{Wx}\,.$$

Here, $\boldsymbol{A}$ defines an orthonormal basis of the space $\mathcal{A}$ and $\boldsymbol{W}$ is a positive definite weight matrix which defines the geometry $\langle \boldsymbol{x}, \boldsymbol{y}\rangle_{\boldsymbol{W}} := \boldsymbol{x}^\top \boldsymbol{Wy}$. When $\boldsymbol{W} = \boldsymbol{I}$, we recover the familiar orthogonal projection. Otherwise, projections can happen 'obliquely' following the geometry defined by $\boldsymbol{W}$.

**Contractions.** A contraction is a linear operator $\boldsymbol{C}$ which contracts all inputs towards the origin:

$$\|\boldsymbol{Cx}\| \le \|\boldsymbol{x}\| \quad \text{for any } \boldsymbol{x}\,.$$

Given these notions, we can rewrite Proposition II as follows.

**Proposition III.** *There exist oblique projection operators $\boldsymbol{P}_1$ and $\boldsymbol{P}_2$, contraction matrices $\boldsymbol{C}_1$ and $\boldsymbol{C}_2$, and orthonormal matrices $\boldsymbol{V}_1$ and $\boldsymbol{V}_2$ such that the model in round $2t$ learned by the alternating knowledge distillation algorithm is*

$$g_{2t}^1(\boldsymbol{x}) = \boldsymbol{l_x}^\top (c\boldsymbol{I} + \boldsymbol{L}_{11})^{-1}\boldsymbol{V}_1^\top \left(\boldsymbol{C}_1\boldsymbol{P}_2^\top \boldsymbol{C}_2\boldsymbol{P}_1^\top\right)^t \boldsymbol{V}_1\boldsymbol{y}^1\,.$$

*In particular, the predictions on agent 2's inputs are*

$$g_{2t}^1(\boldsymbol{X}^2) = \boldsymbol{V}_2^\top \boldsymbol{P}_1^\top \left(\boldsymbol{C}_1\boldsymbol{P}_2^\top \boldsymbol{C}_2\boldsymbol{P}_1^\top\right)^t \boldsymbol{V}_1\boldsymbol{y}^1\,.$$

*Further, the projection matrices satisfy $\boldsymbol{P}_1^\top \boldsymbol{P}_2^\top \boldsymbol{x} = \boldsymbol{x}$ only if $\boldsymbol{x} = 0$.*

The term $\left(\boldsymbol{C}_1\boldsymbol{P}_2^\top \boldsymbol{C}_2\boldsymbol{P}_1^\top\right)^t$ is the only one which depends on $t$ and so captures the dynamics of the algorithm. Two rounds of AKD (first to agent 1 and then back to 2) correspond to a multiplication with $\boldsymbol{C}_1\boldsymbol{P}_2^\top \boldsymbol{C}_2\boldsymbol{P}_1^\top$ i.e. *alternating projections* interspersed by contractions. Thus, the dynamics of AKD is exactly the same as that of alternating projections interspersed by contractions. The projections $\boldsymbol{P}_1^\top$ and $\boldsymbol{P}_2^\top$ have orthogonal ranges whose intersection only contains the origin 0. As Fig. 1b shows, non-orthogonal alternating projections between orthogonal spaces converge to the origin. Contractions further pull the inputs closer to the origin, speeding up the convergence.

**Remark 1.** *To understand the connection to projection more intuitively, suppose that we had a 4-way classification task with agent 1 having data only of the first two classes, and agent 2 with the last two classes. Any model trained by agent 1 will only learn about the first two classes and its predictions on the last two classes will be meaningless. Thus, no information can be transferred between the agents in this setting. More generally, the transfer of knowledge from agent 1 to agent 2 is mediated by its data $\boldsymbol{X}_2$. This corresponds to a* projection *of the knowledge of agent 1 onto the data of agent 2. If there is a mismatch between the two, knowledge is bound to be lost.*

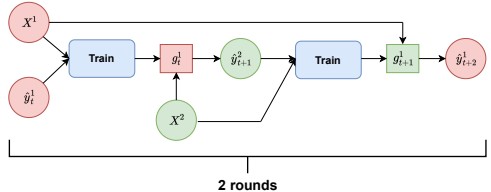

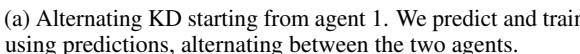

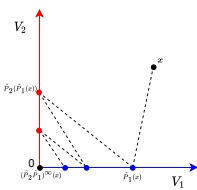

(a) Alternating KD starting from agent 1. We predict and train using predictions, alternating between the two agents.

(b) Alternating oblique projections starting from $x$ converges to 0.

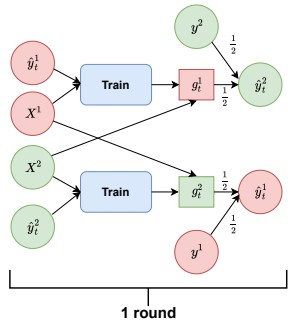

(a) AvgKD scheme

(b) Intuition behind ensemble scheme. In each round, AKD alternates between overfitting the data of agent 1 or of agent 2. We can construct an ensemble out of these models to correct for this bias and quickly converge to the true optima.

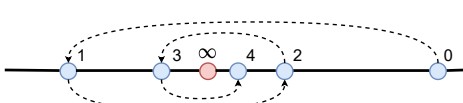

**Speed of degradation.** The alternating projection algorithm converges to the intersection of the subspaces corresponding to the projection operators (their range) (Boyd & Dattorro, 2003). In our particular case, the fixed point of $\boldsymbol{P}_1^\top$ and $\boldsymbol{P}_2^\top$ is 0, and hence this is the point the algorithm will converge to. The contraction operations $\boldsymbol{C}_1$ and $\boldsymbol{C}_2$ only speed up the convergence to the origin 0 (also see Fig. 1b). This explains the degradation process notes in Proposition II. We can go further and examine the rate of degradation using known analyses of alternating projections (Aronszajn, 1950).

**Proposition IV** (Informal). *The rate of convergence to $g_t^1(\boldsymbol{x})$ to 0 gets faster if:*

- *a stronger inductive bias is induced via a larger regularization constant $c$,*
- *the kernels $k_1(\boldsymbol{x}, \boldsymbol{y})$ and $k_2(\boldsymbol{x}, \boldsymbol{y})$ are very different, or*
- *the difference between the datasets $\mathcal{D}_1$ and $\mathcal{D}_2$ as measured by $k_1(\boldsymbol{x}, \boldsymbol{y})$ increases.*

In summary, both data and model heterogeneity may speed up the degradation defeating the purpose of model agnostic FL. All formal proofs and theorem statements are moved to the Appendix.

## 5 ADDITIONAL VARIANTS

In the previous section, we saw that the alternating knowledge distillation (AKD) algorithm over multiple iterations suffered slow degradation, eventually losing all information about the training data. In this section, we explore some alternative approaches which attempt to correct this. We first analyze a simple way to re-inject the training data after every KD iteration which we call averaged distillation. Then, we show an ensemble algorithm that can recover the optimal model $g^{1\star}$.

### 5.1 AVERAGED KNOWLEDGE DISTILLATION

As we saw earlier, each step of knowledge distillation seems to lose some information about the training data, replacing it with the inductive bias of the model. One approach to counter this slow loss of information is to recombine it with the original training data labels such as is commonly done in co-distillation (Sodhani et al., 2020).

**Algorithm.** Recall that agent 1 has data $\mathcal{D}_1 = (\boldsymbol{X}^1, \boldsymbol{y}^1)$ and correspondingly agent 2 has data $\mathcal{D}^2 = (\boldsymbol{X}^2, \boldsymbol{y}^2)$. Now starting from $\hat{\boldsymbol{y}}_0^1 = \boldsymbol{y}^1, \hat{\boldsymbol{y}}_0^2 = \boldsymbol{y}^2$, in each round $t \geq 0$:

   a. Agents 1 and 2 train their model on datasets $(\boldsymbol{X}^1, \hat{\boldsymbol{y}}_t^1)$ and $(\boldsymbol{X}^2, \hat{\boldsymbol{y}}_t^2)$ to obtain $g_t^1$ and $g_t^2$.

    b. Agents exchange their models $g_t^1$ and $g_t^2$ between each other.

    c. Agents use exchanged models to predict labels $\hat{\boldsymbol{y}}_{t+1}^1 = \frac{\boldsymbol{y}^1 + g_t^2(\boldsymbol{X}^1)}{2}$, $\hat{\boldsymbol{y}}_{t+1}^2 = \frac{\boldsymbol{y}^2 + g_t^1(\boldsymbol{X}^2)}{2}$.

The summary the algorithm is depicted in Figure 2a. Again, notice that there is no exchange of raw data but only of the trained models. The main difference between AKD and AvgKD (averaged knowledge distillation) is that we average the predictions with the original labels. This re-injects information $\boldsymbol{y}^1$ and $\boldsymbol{y}^2$ at every iteration, preventing degradation. We theoretically characterize its dynamics next in terms of the afore-mentioned contraction and projection operators.

**Proposition V.** *There exist oblique projection operators $\boldsymbol{P}_1$ and $\boldsymbol{P}_2$, contraction matrices $\boldsymbol{C}_1$ and $\boldsymbol{C}_2$, and orthonormal matrices $\boldsymbol{V}_1$ and $\boldsymbol{V}_2$ such that the model of agent 1 in round $t$ learned by the averaged knowledge distillation (AvgKD) algorithm is*

$$g_t^1(\boldsymbol{x}) = \frac{\boldsymbol{F}}{2}\Big(\sum_{i=0}^{t-1}\Big(\frac{\boldsymbol{C}_1\boldsymbol{P}_2^\top\boldsymbol{C}_2\boldsymbol{P}_1^\top}{4}\Big)^i\Big)\boldsymbol{z}_1 + \frac{\boldsymbol{F}\boldsymbol{C}_1\boldsymbol{P}_2^\top}{4}\Big(\sum_{i=0}^{t-2}\Big(\frac{\boldsymbol{C}_2\boldsymbol{P}_1^\top\boldsymbol{C}_1\boldsymbol{P}_2^\top}{4}\Big)^i\Big)\boldsymbol{z}_2\,.$$

*where $\boldsymbol{F} = \boldsymbol{l}_{\boldsymbol{x}}^\top(c\boldsymbol{I} + \boldsymbol{L}_{11})^{-1}\boldsymbol{V}_1^\top$. Further, in the limit of rounds for any fixed $\boldsymbol{x}$ we have*

$$\lim_{t\to\infty} g_t^1(\boldsymbol{x}) = \frac{\boldsymbol{F}}{2}\Big(\boldsymbol{I} - \frac{\boldsymbol{C}_1\boldsymbol{P}_2^\top\boldsymbol{C}_2\boldsymbol{P}_1^\top}{4}\Big)^\dagger\boldsymbol{z}_1 + \frac{\boldsymbol{F}\boldsymbol{C}_1\boldsymbol{P}_2^\top}{4}\Big(\boldsymbol{I} - \frac{\boldsymbol{C}_2\boldsymbol{P}_1^\top\boldsymbol{C}_1\boldsymbol{P}_2^\top}{4}\Big)^\dagger\boldsymbol{z}_2\,.$$

This shows that the model learned through AvgKD does not degrade to 0, unlike AKD. Instead, it converges to a limit for which we can derive closed-form expressions. Unfortunately, this limit model is still not the same as our desired optimal model $g^{1\star}$. We next try to overcome this using ensembling.

## 5.2 Ensembled Knowledge Distillation

We first analyze how the limit solution of AvgKD differs from the actual optimum $g^{1\star}$. We will build upon this understanding to construct an ensemble that approximates $g^{1\star}$. For simplicity, we assume that the regularization coefficient $c = 0$.

*Understanding AvgKD.* Consider AvgKD algorithm, then

**Proposition VI.** *There exist matrices $\boldsymbol{A}_1$ and $\boldsymbol{A}_2$ such that the minimizer $g^{1\star}$ of objective (5) predicts $g^{1\star}(\boldsymbol{X}_i) = \boldsymbol{A}_1\boldsymbol{\beta}_1 + \boldsymbol{A}_2\boldsymbol{\beta}_2$ for $i \in \{1, 2\}$, where $\boldsymbol{\beta}_1$ and $\boldsymbol{\beta}_2$ satisfy*

$$\begin{pmatrix}\boldsymbol{L}_{11} & \boldsymbol{M}_{12} \\ \boldsymbol{L}_{21} & \boldsymbol{M}_{22}\end{pmatrix}\begin{pmatrix}\boldsymbol{\beta}_1 \\ \boldsymbol{\beta}_2\end{pmatrix} = \begin{pmatrix}\boldsymbol{y}_1 \\ \boldsymbol{y}_2\end{pmatrix}. \tag{7}$$

*In contrast, for the same matrices $\boldsymbol{A}_1$ and $\boldsymbol{A}_2$, the limit models of AvgKD predict $g_\infty^1(\boldsymbol{X}_i) = \frac{1}{2}\boldsymbol{A}_1\boldsymbol{\beta}_1$ and $g_\infty^2(\boldsymbol{X}_i) = -\frac{1}{2}\boldsymbol{A}_2\boldsymbol{\beta}_2$, for $i \in \{1, 2\}$ for $\boldsymbol{\beta}_1$ and $\boldsymbol{\beta}_2$ satisfying*

$$\begin{pmatrix}\boldsymbol{L}_{11} & \frac{\boldsymbol{M}_{12}}{2} \\ \frac{\boldsymbol{L}_{21}}{2} & \boldsymbol{M}_{22}\end{pmatrix}\begin{pmatrix}\boldsymbol{\beta}_1 \\ \boldsymbol{\beta}_2\end{pmatrix} = \begin{pmatrix}\boldsymbol{y}_1 \\ -\boldsymbol{y}_2\end{pmatrix}. \tag{8}$$

By comparing the equations (7) and (8), the output $2(g_\infty^1(\boldsymbol{x}) - g_\infty^2(\boldsymbol{x}))$ is close to the output of $g^{1\star}(\boldsymbol{x})$, except that the off-diagonal matrices are scaled by $\frac{1}{2}$ and we have $-\boldsymbol{y}_2$ on the right hand side. We need two tricks if we want to approximate $g^{1\star}$: first we need an ensemble using differences of models, and second we need to additionally correct for the bias in the algorithm.

*Correcting bias using infinite ensembles.* Consider the initial AKD (alternating knowledge distillation) algorithm illustrated in the Fig. 1a. Let us run two simultaneous runs, ones starting from agent 1 and another starting from agent 2, outputting models $\{g_t^1\}$, and $\{g_t^2\}$ respectively. Then, instead of just using the final models, we construct the following infinite ensemble. For an input $\boldsymbol{x}$, we output:

$$f_\infty(\boldsymbol{x}) = \sum_{t=0}^\infty (-1)^t (g_t^1(\boldsymbol{x}) + g_t^2(\boldsymbol{x})) \tag{9}$$

That is, we take the models from odd steps $t$ with positive signs and from even ones with negative signs and sum their predictions. We call this scheme Ensembled Knowledge Distillation (EKD). The intuition behind our ensemble method is schematically visualized in 1-dimensional case in the Fig. 2b where numbers denote the $t$ variable in the equation (9). We start from the sum of both agents models obtained after learning from ground truth labels (0-th round). Then we subtract the sum of

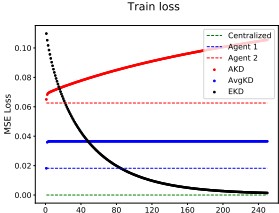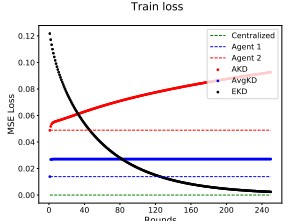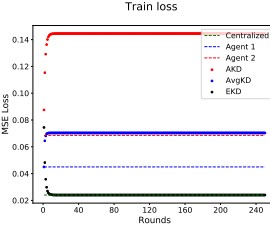

Figure 3: AKD, AvgKD, and EKD methods for linear regression on synthetic data with same data (left), different data (middle), and strong regularization (right). EKD (black) eventually matches centralized performance (dashed green), whereas AvgKD (solid blue) is worse than only local training (dashed blue and red). AKD (in solid red) performs the worst and degrades with increasing rounds.

both agents models obtained after the first round of KD. Then we add the sum of both agents models obtained after the second round of KD and so on. From the section 4.2 we know that in the AKD process with regularization model gradually degrades towards $\mathbf{0}$, in other words, intuitively each next round obtained model adds less value to the whole sum in the EKD scheme. Although, in the lack of regularization models degradation towards $\mathbf{0}$ is not always the case (see App. C). But under such an assumption we we gradually converge to the limit model, which is the point $\infty$ in the Fig. 2b. We formalize this and prove the following.

**Proposition VII.** *The predictions of $f_\infty$ using* (9) *satisfies $f_\infty(\boldsymbol{X}_i) = g^{1^\star}(\boldsymbol{X}_i)$ for $i \in \{1, 2\}$.*

Thus, not only did we succeed in preventing degeneracy to 0, but we also managed to recover the predictions of the optimal model. However, note that this comes at a cost of an infinite ensemble. While we can approximate this using just a finite set of models (as we explore experimentally next), this still does not recover a *single* model which matches $g^{1^\star}$.

## 6 EXPERIMENTS

### 6.1 SETUP

We consider three settings corresponding to the cases Proposition IV with the agents having

- the same model architecture and close data distributions (Same model, Same data)
- different model architectures and close data distributions (Different model, Same data)
- the same model architecture and different data distributions (Same model, Different data).

The toy experiments solve a linear regression problem of the form $\boldsymbol{Ax}^\star = \boldsymbol{b}$. The data $\boldsymbol{A}$ and $\boldsymbol{b}$ is split between the two agents randomly in the 'same data' case, whereas in the 'different data' case the data is sorted according to $b$ before splitting to maximize heterogeneity.

The real world experiments are conducted using Convolutional Neural Network (CNN), Multi-Layer Perceptron network (MLP), and Random Forest (RF). We use squared loss since it is closer to the theoretical setting. In the 'same model' setting both agents use the CNN model, whereas agent 2 instead uses an MLP in the 'different model' setting. Further, we split the training data randomly in the 'same data' setting. For the 'different data' setting, we split the data by labels and take some portion *Alpha* of data from each agent and randomly shuffle taken points between agents. By varying hyperparameter *Alpha* we control the level of data heterogeneity between agents. Notice that if *Alpha* = 0 then datasets are completely different between agents, if *Alpha* = 1 then we have the 'same data' setting with i.i.d. split of data between two agents. All other details are presented in the Appendix A. We next summarize and discuss our results.

### 6.2 RESULTS AND DISCUSSION

**AvgKD > AKD.** In all settings (both synthetic in Fig. 3 and real world in Fig. 4), we see that with the increasing number of rounds the performance of AKD significantly degrades whereas that of AvgKD stabilizes (Fig. 5) regardless of regularization, model and data heterogeneity. Moreover, from the experiments on MNIST in Fig. 4, we see the following: AKD degrades faster if there is regularization used, model heterogeneity or data heterogeneity between agents, and the last plays the most significant role. AvgKD, on the other hand, quickly converges in a few rounds and does not degrade. However, it fails to match the centralized accuracy as well.

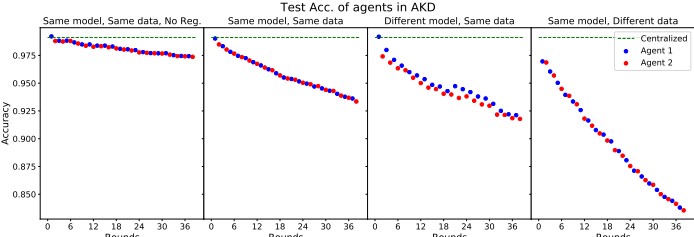

Figure 4: Test accuracy of centralized (dashed green), and AKD on MNIST using model starting from agent 1 (blue) and agent 2 (red) with varying amount of regularization, model heterogeneity, and data heterogeneity. In all cases, performance degrades with increasing rounds with degradation speeding up with the increase in regularization, model heterogeneity, or data heterogeneity.

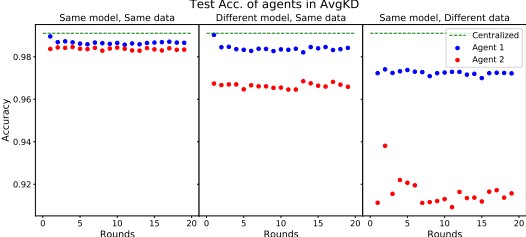

Figure 5: Test accuracy of AvgKD on MNIST using model starting from agent 1 (blue) and agent 2 (red) with varying model heterogeneity, and data heterogeneity. During training regularization is used. In all cases, there is no degradation of performance, though the best accuracy is obtained by agent 1 in round 1 with only local training.

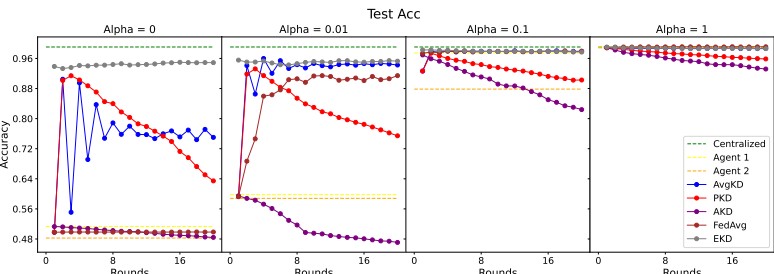

Figure 6: Test accuracy on MNIST with varying data heterogeneity in the setting of 'same model'. In case of high data heterogeneity (small *Alphas*), both agents benefit from the AvgKD, PKD and EKD schemes. Moreover, AvgKD and EKD consistently outperform FedAvg scheme.

**EKD works but needs large ensembles.** In the synthetic setting (Fig. 3), in 250 rounds (ensemble of 500 models) it even matches the centralized model. However, in the real world setting (Fig. 6) the improvement is slower and it does not match centralized performance. This might be due to the small number of rounds run (only 20). EKD is also the only method that improves with subsequent rounds. Finally, we observed that increasing regularization actually speeds up the convergence of EKD in the synthetic setting (Fig. 3).

**Data heterogeneity is the main bottleneck.** In all our experiments, both data and model heterogeneity degraded the performance of AKD, PKD (Parallel KD introduced in App. E) and AvgKD. However, data heterogeneity has a much stronger effect. This confirms our theory that mismatch between agents data leads to loss of information when using knowledge distillation. Overcoming this data heterogeneity is the crucial challenge for practical model agnostic FL. Fig. 6 shows how all schemes behave in dependence on data heterogeneity. Indeed, the higher data heterogeneity the faster speed of degradation for the AKD and PKD schemes. In the case of the AvgKD scheme, we see that agents do improve over their local models and this improvement is larger with more data heterogeneity.

### 6.3 ADDITIONAL EXTENSIONS AND EXPERIMENTS

In App. G, we extend our algorithms to $M$ agents and show experimentally in App. H.5 for AvgKD algorithm that the same trends hold there as well. Our conclusions also hold for the cross-entropy loss (Figs. 11, 12), for highly heterogeneous model case with MLP and Random Forests (Figs. 13, 14), as well on other datasets and models (VGG on CIFAR10 in Figs. 9, 10). In the latter, we see the same trends as on MNIST for all the schemes except EKD which is probably due to the use of cross-entropy loss function and, as a result, models being further from the kernel regime. Moreover, speed of degradation is higher if there is the model heterogeneity (Figs. 8, 10) and EKD does not help even on MNIST dataset (Fig. 8). Finally, all the schemes are compared to the more standard FedAvg that is not applicable in the 'different model' setting and is outperformed by the AvgKD scheme at highly data heterogeneous regimes. That is, AvgKD consistently outperforms all the methods at highly data heterogeneous regimes indicating it is the most promising variant.

## 7 CONCLUSION

While the stochastic optimization framework has been very useful in analyzing and developing new algorithms for federated learning so far, it fundamentally cannot capture learning with different models. We instead introduced the federated kernel regression framework where we formalized notions of both model and data heterogeneity. Using this, we analyzed different knowledge distillation schemes and came to the conclusion that data heterogeneity poses a fundamental challenge limiting the knowledge that can be transmitted. Further, these theoretical predictions were exactly reflected in our deep learning experiments as well. Overcoming this data heterogeneity will be crucial to making KD based model agnostic federated algorithms practical.

We also utilized our framework to design a novel ensembling method motivated by our theory. However, this method could require very large ensembles (up to 500 models) in order to match the centralized performance. Thus, we view our method not as a practical algorithm, but more of a demo on how our framework can be leveraged. Similarly, our experiments are preliminary and are not on real world complex datasets. We believe there is great potential in further exploring our results and framework, especially in investigating how to mitigate the effect of data heterogeneity in knowledge distillation.

## ACKNOWLEDGEMENTS

We are really grateful to Martin Jaggi for his insightful comments and support throughout this work. SPK is partly funded by an SNSF Fellowship and AA is funded by a research scholarship from MLO lab, EPFL headed by Martin Jaggi. SPK also thanks Celestine Dünner for conversations inspiring this project.

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

## A  Experiments details

In all real world experiments we use the Adam optimizer with a default regularization (weight decay) of $3 \times 10^{-4}$, unless in the 'no regularization' case when it is set to 0. We split the data between 2 agents by giving a bigger part of data to agent 1 at all 'same data' experiments. That is, for 'different data' experiments and heterogeneous data experiments where we vary *Alpha* hyperparameter we split data equally between 2 agents. The non-equal split can help us to see another effect in the experiments: if agent 2 (with less data) benefits from the communication with agent 1 (with more data). In heterogeneous data experiments, we explore the significance of data heterogeneity effect on the behavior of KD scheme. That is why we design an almost ideal setting at all such experiments: the 'same model' setting (except experiments with RF and MLP), the equal split of data in terms of its amount between agents.

**Toy experiments.**  The toy experiments solve a linear regression problem of the form $\boldsymbol{A}\boldsymbol{x}^{\star} = \boldsymbol{b}$ where $\boldsymbol{A} \in \mathbb{R}^{n \times d}$ and $\boldsymbol{x}^{\star} \in \mathbb{R}^{d}$ is randomly generated for $n = 1.5d$ and $d = 100$. Then, the data $\boldsymbol{A}$ and $\boldsymbol{b}$ is split between the two agents at proportion $0.6/0.4$. This is done randomly in the 'same data' case, whereas in the 'different data' case the data is sorted according to $b$ before splitting to maximize heterogeneity. This experiment with the linear kernel is supposed to show if our theory is correct, especially if the EKD scheme really works.

**Real world experiments.**  The real world experiments are conducted using CNN and MLP networks on MNIST, MLP network and RF model on MNIST, and VGG16[1] (Simonyan & Zisserman, 2015) and CNN models on CIFAR10 datasets. Further, we split the training data randomly at proportion $0.7/0.3$ in the 'same data' setting. For the 'different data' setting, we split the data by labels: agent 1 has '0' to '4' labeled data points, agent 2 has '5' to '9'. Then we take randomly from each agent some *Alpha* $= 0.1$ portion of data, combine it and randomly return data points to both agents from this combined dataset.

## B  Alternating KD with regularization

**Different models.**  Keeping the notation of section 4 we can construct the following matrix:

$$\boldsymbol{K} = \begin{pmatrix} \boldsymbol{L} & 0 \\ 0 & \boldsymbol{M} \end{pmatrix} = \begin{pmatrix} \boldsymbol{L}_{11} & \boldsymbol{L}_{12} & 0 & 0 \\ \boldsymbol{L}_{21} & \boldsymbol{L}_{22} & 0 & 0 \\ 0 & 0 & \boldsymbol{M}_{11} & \boldsymbol{M}_{12} \\ 0 & 0 & \boldsymbol{M}_{21} & \boldsymbol{M}_{22} \end{pmatrix}.$$

Notice that each of the sub-blocks $\boldsymbol{L}$ and $\boldsymbol{M}$ are symmetric positive semi-definite matrices. This makes matrix $\boldsymbol{K}$ symmetric positive semi-definite matrix and it has eigendecomposition form:

$$\boldsymbol{K} = \boldsymbol{V}^{\top} \boldsymbol{D} \boldsymbol{V} \quad \text{and} \quad \boldsymbol{V} = \left( \boldsymbol{V}_1 \boldsymbol{V}_2 \tilde{\boldsymbol{V}}_1 \tilde{\boldsymbol{V}}_2 \right),$$

where $\boldsymbol{D}$ and $\boldsymbol{V}$ are $\mathbb{R}^{2(N_1+N_2) \times 2(N_1+N_2)}$ diagonal and orthogonal matrices correspondingly. This means that the $\mathbb{R}^{2(N_1+N_2) \times 2(N_1+N_2)}$ matrices $\boldsymbol{V}_1, \boldsymbol{V}_2, \tilde{\boldsymbol{V}}_1, \tilde{\boldsymbol{V}}_2$ are also orthogonal to each other. Then one can deduce:

$$\boldsymbol{L}_{ij} = \boldsymbol{V}_i^{\top} \boldsymbol{D} \boldsymbol{V}_j, \quad \text{and} \quad \boldsymbol{M}_{ij} = \tilde{\boldsymbol{V}}_i^{\top} \boldsymbol{D} \tilde{\boldsymbol{V}}_j \quad \forall i, j = 1, 2.$$

In AKD setting only agent 1 has labeled data. The solution of learning from the dataset $\mathcal{D}_1$ evaluated at set $\mathcal{X}_2$ is the following:

$$g_0^1(\mathcal{X}_2) = \boldsymbol{L}_{21}(c\boldsymbol{I} + \boldsymbol{L}_{11})^{-1}\boldsymbol{y}_1 = \boldsymbol{V}_2^{\top}((\boldsymbol{V}_1^{\top}(c\boldsymbol{I} + \boldsymbol{D})\boldsymbol{V}_1)^{-1}\boldsymbol{V}_1^{\top}\boldsymbol{D})^{\top}\boldsymbol{y}_1. \quad (10)$$

Let us introduce notation:

$$\boldsymbol{P}_1 = \boldsymbol{V}_1(\boldsymbol{V}_1^{\top}(c\boldsymbol{I} + \boldsymbol{D})\boldsymbol{V}_1)^{-1}\boldsymbol{V}_1^{\top}(c\boldsymbol{I} + \boldsymbol{D}) \quad \text{and} \quad \tilde{\boldsymbol{P}}_2 = \tilde{\boldsymbol{V}}_2(\tilde{\boldsymbol{V}}_2^{\top}(c\boldsymbol{I} + \boldsymbol{D})\tilde{\boldsymbol{V}}_2)^{-1}\tilde{\boldsymbol{V}}_2^{\top}(c\boldsymbol{I} + \boldsymbol{D}).$$

These are well known oblique (weighted) projection matrices on the subspaces spanned by the columns of matrices $\boldsymbol{V}_1$ and $\tilde{\boldsymbol{V}}_2$ correspondingly, where the scalar product is defined with Gram matrix $\boldsymbol{G} = c\boldsymbol{I} + \boldsymbol{D}$. Similarly one can define $\boldsymbol{P}_2$ and $\tilde{\boldsymbol{P}}_1$.

---

[1]This model is also a convolutional neural network, but in our experiments it is bigger than CNN model.

Given the introduced notation and using the fact that $V_2^\top V_1 = 0$, we can rewrite equation 10 as:

$$g_0^1(\mathcal{X}_2) = V_2^\top P_1^\top V_1 y_1 \,. \tag{11}$$

Similarly, given $\mathcal{X}_1 \subseteq \mathcal{D}_1$ and agent 1 learned from $\hat{\mathcal{Y}}_2 = g_0^1(\mathcal{X}_2)$, inferred prediction $\hat{\mathcal{Y}}_1 = g_1^1(\mathcal{X}_1)$ by agent 2 can be written as:

$$g_1^1(\mathcal{X}_1) = \tilde{V}_1^\top \tilde{P}_2^\top \tilde{V}_2 V_2^\top P_1^\top z_1, \quad \text{where} \quad z_1 = V_1 y_1 \,.$$

At this point we need to introduce additional notation:

$$C_1 = V_1 \tilde{V}_1^\top \quad \text{and} \quad \tilde{C}_2 = \tilde{V}_2 V_2^\top \,.$$

These are matrices of contraction linear mappings.

One can now repeat the whole process again with replacement $\mathcal{Y}_1$ by $\hat{\mathcal{Y}}_1$. The predictions by agent 1 and agent 2 after such $2t$ rounds of AKD are:

$$g_{2t}^1(\mathcal{X}_2) = V_2^\top P_1^\top \left( C_1 \tilde{P}_2^\top \tilde{C}_2 P_1^\top \right)^t z_1 \quad \text{and} \quad g_{2t+1}^1(\mathcal{X}_1) = \tilde{V}_1^\top \tilde{P}_2^\top \tilde{C}_2 P_1^\top \left( C_1 \tilde{P}_2^\top \tilde{C}_2 P_1^\top \right)^t z_1 \,. \tag{12}$$

If one considers the case where agents have the same kernel $u_1 = u_2 = u$ then one should 'remove all tildas' in the above expressions and the obtained expressions are applied. This means $M = L$, $V_1 = \tilde{V}_1$ and $V_2 = \tilde{V}_2$. Crucial changes in this case are $C_1 = \tilde{C}_2 \to I$ and $\left( C_1 \tilde{P}_2^\top \tilde{C}_2 P_1^\top \right)^t \to \left( P_2^\top P_1^\top \right)^t$. The matrix $\left( P_2^\top P_1^\top \right)^t$ is an alternating projection (Boyd & Dattorro, 2003) operator after $t$ steps. Given 2 closed convex sets, alternating projection algorithm in the limit finds a point in the intersection of these sets, provided they intersect (Boyd & Dattorro, 2003; Lewis et al., 2007). In our case matrices operators $P_1$ and $P_2$ project onto linear spaces spanned by columns of matrices $V_1$ and $V_2$ correspondingly. These are orthogonal linear subspaces, hence the unique point of their intersection is the origin point $0$. This means that $\left( P_2^\top P_1^\top \right)^t \xrightarrow{t \to \infty} 0$ and in the limit of such AKD procedure both agents predict 0 for any data point.

**Speed of degradation.** The speed of convergence for the alternating projections algorithm is known and defined by the minimal angle $\phi$ between corresponding sets (Aronszajn, 1950) (only non-zero elements from sets one has to consider):

$$\| (P_2 P_1)^t v \| \le (\cos(\phi))^{2t-1} \|v\| = (\cos(\phi))^{2t-1} \quad \text{as} \quad \|v\| = 1 \,,$$

where $v$ is one of the columns of matrix $V_1$. In our case we can write the expression for the cosine:

$$\cos(\phi) = \max_{v_1, v_2}\left( \frac{|v_1^\top (cI + D) v_2|}{\sqrt{(v_1^\top (cI + D) v_1) \cdot (v_2^\top (cI + D) v_2)}} \right) = \max_{v_1, v_2}\left( \frac{|v_1^\top D v_2|}{\sqrt{(c + v_1^\top D v_1) \cdot (c + v_2^\top D v_2)}} \right) \,.$$

where $v_1$ and $v_2$ are the vectors from subspaces spanned by columns of matrices $V_1$ and $V_2$ correspondingly. Hence the speed of convergence depends on the elements of the matrix $L_{12}$. Intuitively these elements play the role of the measure of 'closeness' between data points. This means that the 'closer' points of sets $\mathcal{X}_1$ and $\mathcal{X}_2$ the higher absolute values of elements in matrix $L_{12}$. Moreover, we see inversely proportional dependence on the regularization constant $c$. All these sums up in the following proposition which is the formal version of the proposition IV:

**Proposition VIII** (Formal). *The rate of convergence of $g_t^1(x)$ to 0 gets faster if:*

- *larger regularization constant $c$ is used during the training,*
- *smaller the eigenvalues of the matrix $V_1 \tilde{V}_1^\top$, or*
- *smaller absolute value of non-diagonal block $L_{12}$*

## C  Alternating KD without regularization

Before we saw that models of both agents degrade if one uses regularization. A natural question to ask if models will degrade in the lack of regularization. Consider the problem (1) with $c \to 0$, so

the regularization term cancels out. In the general case, there are many possible solutions as kernel matrix $\boldsymbol{K}$ may have $0$ eigenvalues. Motivated by the fact that Stochastic Gradient Descent (SGD) tends to find the solution with minimal norm (Wilson et al., 2017), we propose to analyze the minimal norm solution in the problem of alternating knowledge distillation with 2 agents each with private dataset. For simplicity let us assume that agents have the same model architecture. Then the agent I solution evaluated at set $\mathcal{X}_2$ with all the above notation can be written as:

$$g_0^\star(\mathcal{X}_2) = \boldsymbol{K}_{21}\boldsymbol{K}_{11}^\dagger \boldsymbol{y}_1 = \boldsymbol{V}_2^\top \boldsymbol{D}\boldsymbol{V}_1(\boldsymbol{V}_1^\top \boldsymbol{D}\boldsymbol{V}_1)^\dagger \boldsymbol{y}_1, \tag{13}$$

where $\dagger$ stands for pseudoinverse.
In this section let us consider 3 possible settings one can have:

1. Self-distillation: The datasets and models of both agents are the same.

2. Distillation with $\boldsymbol{K} > 0$: The datasets of both agents are different and private. Kernel matrix $\boldsymbol{K}$ is positive definite.

3. Distillation with $\boldsymbol{K} \geq 0$: The datasets of both agents are different and private. Kernel matrix $\boldsymbol{K}$ is positive semi-definite.

**Self-distillation.** Given the dataset $\mathcal{D} = \mathcal{X} \times \mathcal{Y}$, the solution of the supervised learning with evaluation at any $\boldsymbol{x} \in \mathbb{R}^d$ is:

$$g_0^\star(\boldsymbol{x}) = \boldsymbol{l}_{\boldsymbol{x}}^\top (\boldsymbol{V}^\top \boldsymbol{D}\boldsymbol{V})^\dagger \boldsymbol{y}.$$

Then the expression for the self-distillation step with evaluation at any $\boldsymbol{x} \in \mathbb{R}^d$ is:

$$g_1^\star(\boldsymbol{x}) = \boldsymbol{l}_{\boldsymbol{x}}^\top (\boldsymbol{V}^\top \boldsymbol{D}\boldsymbol{V})^\dagger \boldsymbol{V}^\top \boldsymbol{D}\boldsymbol{V}(\boldsymbol{V}^\top \boldsymbol{D}\boldsymbol{V})^\dagger \boldsymbol{y} = \boldsymbol{l}_{\boldsymbol{x}}^\top (\boldsymbol{V}^\top \boldsymbol{D}\boldsymbol{V})^\dagger \boldsymbol{y} = g_0^\star(\boldsymbol{x}),$$

where property of pseudoinverse matrix was used: $\boldsymbol{A}^\dagger \boldsymbol{A}\boldsymbol{A}^\dagger = \boldsymbol{A}^\dagger$.
That is, self-distillation round does not change obtained model. One can repeat self-distillation step $t$ times and there is no change in the model:

$$g_t^\star(\boldsymbol{x}) = \boldsymbol{l}_{\boldsymbol{x}}^\top (\boldsymbol{V}^\top \boldsymbol{D}\boldsymbol{V})^\dagger \boldsymbol{y} = g_0^\star(\boldsymbol{x}),$$

Hence in the no regularization setting self-distillation step does not give any change in the obtained model.

**Distillation with $\boldsymbol{K} > 0$.** In this setting we have the following identity $\boldsymbol{K}^\dagger = \boldsymbol{K}^{-1}$ and results are quite similar to the setting with regularization. But now the Gram matrix of scalar product in linear space is $\boldsymbol{D}$ instead of $c\boldsymbol{I} + \boldsymbol{D}$ in the regularized setting. By assumption on matrix $\boldsymbol{K}$ it follows that $\boldsymbol{D}$ is of full rank and the alternating projection algorithm converges to the origin point $\boldsymbol{0}$. Therefore in the limit of AKD steps predictions by models of two agents will degrade towards $\boldsymbol{0}$.

**Distillation with $\boldsymbol{K} \geq 0$.** In this setting kernel matrix $\boldsymbol{K}$ has at least one $0$ eigenvalue and we take for the analysis the minimal norm solution (13). We can rewrite this solution:

$$g_0^\star(\mathcal{X}_2) = \boldsymbol{V}_2^\top \boldsymbol{D}\boldsymbol{V}_1(\boldsymbol{V}_1^\top \boldsymbol{D}\boldsymbol{V}_1)^\dagger \boldsymbol{y}_1 = \boldsymbol{V}_2^\top \hat{\boldsymbol{P}}_1^\top \boldsymbol{z}_1,$$

where $\hat{\boldsymbol{P}}_1 = \boldsymbol{V}_1(\boldsymbol{V}_1^\top \boldsymbol{D}\boldsymbol{V}_1)^\dagger \boldsymbol{V}_1^\top \boldsymbol{D}$ and $\boldsymbol{z}_1 = \boldsymbol{V}_1\boldsymbol{y}_1$.

One can notice the following projection properties of matrix $\hat{\boldsymbol{P}}_1$:

$$\hat{\boldsymbol{P}}_1^2 = \hat{\boldsymbol{P}}_1 \quad \text{and} \quad \hat{\boldsymbol{P}}_1\boldsymbol{V}_1(\boldsymbol{V}_1^\top \boldsymbol{D}\boldsymbol{V}_1)^\dagger = \boldsymbol{V}_1(\boldsymbol{V}_1^\top \boldsymbol{D}\boldsymbol{V}_1)^\dagger.$$

That is, $\hat{\boldsymbol{P}}_1$ is a projection matrix with eigenspace spanned by columns of the matrix $\boldsymbol{V}_1(\boldsymbol{V}_1^\top \boldsymbol{D}\boldsymbol{V}_1)^\dagger$. Similarly, one can define $\hat{\boldsymbol{P}}_2$. Then the solution for the first AKD step evaluated at set $\mathcal{X}_1$ is:

$$g_1^\star(\mathcal{X}_1) = \boldsymbol{V}_1^\top \boldsymbol{D}\boldsymbol{V}_2(\boldsymbol{V}_2^\top \boldsymbol{D}\boldsymbol{V}_2)^\dagger \boldsymbol{V}_2^\top \hat{\boldsymbol{P}}_1^\top \boldsymbol{z}_1 = \boldsymbol{V}_1^\top \hat{\boldsymbol{P}}_2^\top \hat{\boldsymbol{P}}_1^\top \boldsymbol{z}_1,$$

and after $t$ rounds we obtain for agent I and agent II correspondingly:

$$g_{2t}^\star(\mathcal{X}_2) = \boldsymbol{V}_2^\top \hat{\boldsymbol{P}}_1^\top (\hat{\boldsymbol{P}}_2^\top \hat{\boldsymbol{P}}_1^\top)^t \boldsymbol{z}_1 \quad \text{and} \quad g_{2t+1}^\star(\mathcal{X}_1) = \boldsymbol{V}_1^\top (\hat{\boldsymbol{P}}_2^\top \hat{\boldsymbol{P}}_1^\top)^{t+1} \boldsymbol{z}_1.$$

In the limit of the distillation rounds operator $(\hat{\boldsymbol{P}}_2\hat{\boldsymbol{P}}_1)^t$ tends to the projection on the intersection of 2 subspaces spanned by the columns of matrices $\boldsymbol{V}_1(\boldsymbol{V}_1^\top \boldsymbol{D}\boldsymbol{V}_1)^\dagger$ and $\boldsymbol{V}_2(\boldsymbol{V}_2^\top \boldsymbol{D}\boldsymbol{V}_2)^\dagger$. We should

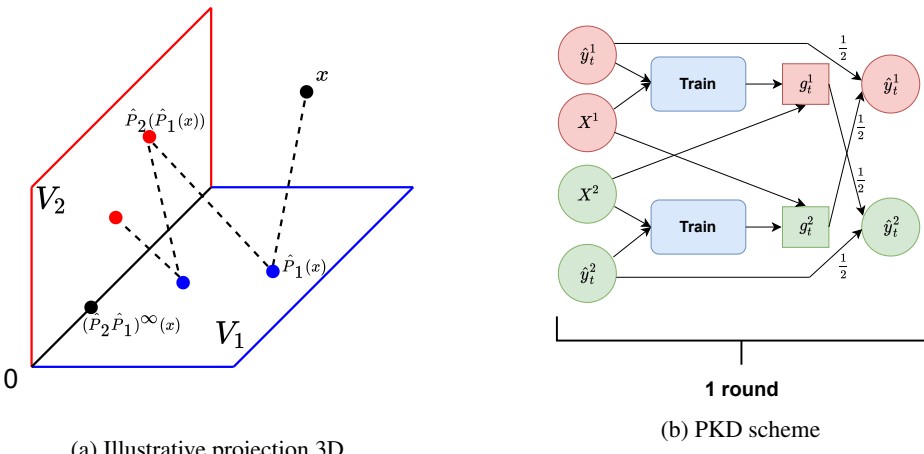

(a) Illustrative projection 3D    (b) PKD scheme

highlight that in general, the intersection set in this case consists not only from the origin point $\mathbf{0}$ but some rays that lie simultaneously in the eigenspaces of both projectors $\hat{P}_1$ and $\hat{P}_2$. The illustrative example is shown in the Fig. 7a. We perform the alternating projection algorithm between orthogonal linear spaces that have a ray in the intersection and we converge to some non-zero point that belongs to this ray. The intersection of eigenspaces of both projectors $\hat{P}_1$ and $\hat{P}_2$ is the set of points $\boldsymbol{x} \in \text{Span}(\boldsymbol{V}_1(\boldsymbol{V}_1^\top \boldsymbol{D} \boldsymbol{V}_1)^\dagger)$ s.t. $\hat{P}_1 \hat{P}_2 \boldsymbol{x} = \boldsymbol{x}$.

## D  AVERAGED KD

In this section we present the detailed analysis of the algorithm presented in the section 4.2 keeping the notation of the section B. As a reminder, models of both agents after round $t$ of AvgKD algorithm are as follows:

$$g_t^1(\boldsymbol{X}_2) = \frac{1}{2}\boldsymbol{L}_{21}(c\boldsymbol{I} + \boldsymbol{L}_{11})^{-1}(\boldsymbol{y}_1 + g_{t-1}^2(\boldsymbol{X}_1)) = \frac{1}{2}\boldsymbol{V}_2^\top \boldsymbol{P}_1^\top (\boldsymbol{z}_1 + g_{t-1}^2),$$

$$g_t^2(\boldsymbol{X}_1) = \frac{1}{2}\boldsymbol{M}_{12}(c\boldsymbol{I} + \boldsymbol{M}_{22})^{-1}(g_{t-1}^1(\boldsymbol{X}_2) + \boldsymbol{y}_2) = \frac{1}{2}\tilde{\boldsymbol{V}}_1^\top \tilde{\boldsymbol{P}}_2^\top (g_{t-1}^1 + \boldsymbol{z}_2).$$

where

$$g_{t-1}^1 = \frac{1}{2}(c\boldsymbol{I} + \boldsymbol{D})\boldsymbol{V}_1(c\boldsymbol{I} + \boldsymbol{L}_{11})^{-1}(\boldsymbol{y}_1 + g_{t-2}^2(\boldsymbol{X}_1)), \quad \forall t \geq 2 \tag{14}$$

$$g_{t-1}^2 = \frac{1}{2}(c\boldsymbol{I} + \boldsymbol{D})\tilde{\boldsymbol{V}}_2(c\boldsymbol{I} + \boldsymbol{M}_{22})^{-1}(g_{t-2}^1(\boldsymbol{X}_2) + \boldsymbol{y}_2), \quad \forall t \geq 2 \tag{15}$$

$$t = 1: \quad g_0^1 = \boldsymbol{P}_1^\top \boldsymbol{z}_1 \quad \text{and} \quad g_0^2 = \tilde{\boldsymbol{P}}_2^\top \boldsymbol{z}_2, \quad \text{where} \quad \boldsymbol{z}_1 = \boldsymbol{V}_1 \boldsymbol{y}^1, \quad \boldsymbol{z}_2 = \tilde{\boldsymbol{V}}_2 \boldsymbol{y}^2. \tag{16}$$

The illustration of this process is presented in the Fig. 2a.

Consider the sequence of solutions for the agent 1 with evaluation at $\boldsymbol{X}_2$:

- Supervised Learning:
$$g_0^1(\boldsymbol{X}_2) = \boldsymbol{V}_2^\top \boldsymbol{P}_1^\top \boldsymbol{z}_1$$

- 1st round of KD:
$$g_1^1(\boldsymbol{X}_2) = \boldsymbol{V}_2^\top \frac{\boldsymbol{P}_1^\top}{2}(\boldsymbol{z}_1 + \boldsymbol{C}_1 \tilde{\boldsymbol{P}}_2^\top \boldsymbol{z}_2)$$

- 2nd round of KD:
$$g_2^1(\boldsymbol{X}_2) = \boldsymbol{V}_2^\top \frac{\boldsymbol{P}_1^\top}{2}(\boldsymbol{z}_1 + \frac{\boldsymbol{C}_1 \tilde{\boldsymbol{P}}_2^\top}{2}\boldsymbol{z}_2 + \frac{\boldsymbol{C}_1 \tilde{\boldsymbol{P}}_2^\top \tilde{\boldsymbol{C}}_2 \boldsymbol{P}_1^\top}{2}\boldsymbol{z}_1)$$

- 3rd round of KD:
$$g_3^1(\boldsymbol{X}_2) = \boldsymbol{V}_2^\top \frac{\boldsymbol{P}_1^\top}{2}(\boldsymbol{z}_1 + \frac{\boldsymbol{C}_1 \tilde{\boldsymbol{P}}_2^\top}{2}\boldsymbol{z}_2 + \frac{\boldsymbol{C}_1 \tilde{\boldsymbol{P}}_2^\top \tilde{\boldsymbol{C}}_2 \boldsymbol{P}_1^\top}{4}\boldsymbol{z}_1 + \frac{\boldsymbol{C}_1 \tilde{\boldsymbol{P}}_2^\top \tilde{\boldsymbol{C}}_2 \boldsymbol{P}_1^\top \boldsymbol{C}_1 \tilde{\boldsymbol{P}}_2^\top}{4}\boldsymbol{z}_2)$$

- $t$-th round of KD:

$$g_t^1(\boldsymbol{X}_2) = \boldsymbol{V}_2^\top \frac{\boldsymbol{P}_1^\top}{2}(\sum_{i=0}^{t}(\frac{\boldsymbol{C}_1\tilde{\boldsymbol{P}}_2^\top\tilde{\boldsymbol{C}}_2\boldsymbol{P}_1^\top}{4})^i)\boldsymbol{z}_1 + \boldsymbol{V}_2^\top \frac{\boldsymbol{P}_1^\top\boldsymbol{C}_1\tilde{\boldsymbol{P}}_2^\top}{4}(\sum_{i=0}^{t-1}(\frac{\tilde{\boldsymbol{C}}_2\boldsymbol{P}_1^\top\boldsymbol{C}_1\tilde{\boldsymbol{P}}_2^\top}{4})^i)\boldsymbol{z}_2$$

- The limit of KD steps:

$$g_\infty^1(\boldsymbol{X}_2) = \boldsymbol{V}_2^\top \frac{\boldsymbol{P}_1^\top}{2}(\boldsymbol{I} - \frac{\boldsymbol{C}_1\tilde{\boldsymbol{P}}_2^\top\tilde{\boldsymbol{C}}_2\boldsymbol{P}_1^\top}{4})^\dagger\boldsymbol{z}_1 + \boldsymbol{V}_2^\top \frac{\boldsymbol{P}_1^\top\boldsymbol{C}_1\tilde{\boldsymbol{P}}_2^\top}{4}(\boldsymbol{I} - \frac{\tilde{\boldsymbol{C}}_2\boldsymbol{P}_1^\top\boldsymbol{C}_1\tilde{\boldsymbol{P}}_2^\top}{4})^\dagger\boldsymbol{z}_2,$$

where † stands for pseudoinverse.

Let us analyze the limit solution and consider the first term of its expression. One can deduce the following identity: [2]

$$\boldsymbol{V}_2^\top \frac{\tilde{\boldsymbol{P}}_1^\top}{2}(\boldsymbol{I} - \frac{\boldsymbol{C}_1\tilde{\boldsymbol{P}}_2^\top\tilde{\boldsymbol{C}}_2\boldsymbol{P}_1^\top}{4})^\dagger\boldsymbol{z}_1 = \tag{17}$$

$$\frac{\boldsymbol{V}_2^\top\boldsymbol{D}\boldsymbol{V}_1}{2}(c\boldsymbol{I} + \boldsymbol{V}_1^\top\boldsymbol{D}\boldsymbol{V}_1 - \frac{\tilde{\boldsymbol{V}}_1^\top\boldsymbol{D}\tilde{\boldsymbol{V}}_2}{2}(c\boldsymbol{I} + \tilde{\boldsymbol{V}}_2^\top\boldsymbol{D}\tilde{\boldsymbol{V}}_2)^{-1}\frac{\boldsymbol{V}_2^\top\boldsymbol{D}\boldsymbol{V}_1}{2})^\dagger\boldsymbol{y}_1 \tag{18}$$

In a similar manner, we can deal with the second term:

$$\boldsymbol{V}_2^\top\frac{\boldsymbol{P}_1^\top\boldsymbol{C}_1\tilde{\boldsymbol{P}}_2^\top}{4}(\boldsymbol{I} - \frac{\tilde{\boldsymbol{C}}_2\boldsymbol{P}_1^\top\boldsymbol{C}_1\tilde{\boldsymbol{P}}_2^\top}{4})^\dagger\boldsymbol{z}_2 =$$

$$\frac{\boldsymbol{V}_2^\top\boldsymbol{D}\boldsymbol{V}_1}{2}(c\boldsymbol{I} + \boldsymbol{V}_1^\top\boldsymbol{D}\boldsymbol{V}_1 - \frac{\tilde{\boldsymbol{V}}_1^\top\boldsymbol{D}\tilde{\boldsymbol{V}}_2}{2}(c\boldsymbol{I} + \tilde{\boldsymbol{V}}_2^\top\boldsymbol{D}\tilde{\boldsymbol{V}}_2)^{-1}\frac{\boldsymbol{V}_2^\top\boldsymbol{D}\boldsymbol{V}_1}{2})^\dagger\frac{\tilde{\boldsymbol{V}}_1^\top\boldsymbol{D}\tilde{\boldsymbol{V}}_2}{2}(c\boldsymbol{I} + \tilde{\boldsymbol{V}}_2^\top\boldsymbol{D}\tilde{\boldsymbol{V}}_2)^{-1}\boldsymbol{y}_2$$

Now, we notice Schur complement expression in the equation (18):

$$(c\boldsymbol{I} + \boldsymbol{V}_1^\top\boldsymbol{D}\boldsymbol{V}_1 - \frac{\tilde{\boldsymbol{V}}_1^\top\boldsymbol{D}\tilde{\boldsymbol{V}}_2}{2}(c\boldsymbol{I} + \tilde{\boldsymbol{V}}_2^\top\boldsymbol{D}\tilde{\boldsymbol{V}}_2)^{-1}\frac{\boldsymbol{V}_2^\top\boldsymbol{D}\boldsymbol{V}_1}{2}).$$

This means that we can consider the following problem:

$$\begin{pmatrix} \boldsymbol{L}_{11} + c\boldsymbol{I} & \frac{\boldsymbol{M}_{12}}{2} \\ \frac{\boldsymbol{L}_{21}}{2} & \boldsymbol{M}_{22} + c\boldsymbol{I} \end{pmatrix}\begin{pmatrix} \boldsymbol{\beta}_1 \\ \boldsymbol{\beta}_2 \end{pmatrix} = \begin{pmatrix} \boldsymbol{V}_1^\top\boldsymbol{D}\boldsymbol{V}_1 + c\boldsymbol{I} & \frac{\tilde{\boldsymbol{V}}_1^\top\boldsymbol{D}\tilde{\boldsymbol{V}}_2}{2} \\ \frac{\boldsymbol{V}_2^\top\boldsymbol{D}\boldsymbol{V}_1}{2} & \tilde{\boldsymbol{V}}_2^\top\boldsymbol{D}\tilde{\boldsymbol{V}}_2 + c\boldsymbol{I} \end{pmatrix}\begin{pmatrix} \boldsymbol{\beta}_1 \\ \boldsymbol{\beta}_2 \end{pmatrix} = \begin{pmatrix} \boldsymbol{y}_1 \\ -\boldsymbol{y}_2 \end{pmatrix}, \tag{19}$$

and derive that $g_\infty^1(\boldsymbol{X}_2) = \frac{\boldsymbol{V}_2^\top\boldsymbol{D}\boldsymbol{V}_1}{2}\boldsymbol{\beta}_1$ and $g_\infty^2(\boldsymbol{X}_1) = -\frac{\tilde{\boldsymbol{V}}_1^\top\boldsymbol{D}\tilde{\boldsymbol{V}}_2}{2}\boldsymbol{\beta}_2$, where $\boldsymbol{\beta}_1, \boldsymbol{\beta}_2$ are defined as the solution to the problem (19). Given this, we can derive the following:

$$\boldsymbol{V}_1^\top\boldsymbol{D}\boldsymbol{V}_1\boldsymbol{\beta}_1 + \frac{\tilde{\boldsymbol{V}}_1^\top\boldsymbol{D}\tilde{\boldsymbol{V}}_2}{2}\boldsymbol{\beta}_2 = \boldsymbol{y}_1 - c\boldsymbol{\beta}_1 = 2g_\infty^1(\mathcal{X}_1) - g_\infty^2(\mathcal{X}_1), \tag{20}$$

$$-\frac{\boldsymbol{V}_2^\top\boldsymbol{D}\boldsymbol{V}_1}{2}\boldsymbol{\beta}_1 - \tilde{\boldsymbol{V}}_2^\top\boldsymbol{D}\tilde{\boldsymbol{V}}_2\boldsymbol{\beta}_2 = \boldsymbol{y}_2 + c\boldsymbol{\beta}_2 = 2g_\infty^2(\boldsymbol{X}_2) - g_\infty^1(\boldsymbol{X}_2). \tag{21}$$

As one can see, there is a strong relation between the limit KD solutions and the solution of a linear system of equations with modified matrix $\boldsymbol{K}$ and right-hand side. Mainly, we take the kernel matrix and divide its non-diagonal blocks by 2, which intuitively shows that our final model accounts for the reduction of the 'closeness' between datasets $\mathcal{D}_1$ and $\mathcal{D}_2$. And on the right-hand side, we see $-\boldsymbol{y}_2$ instead of $\boldsymbol{y}_2$ which is quite a 'artificial' effect. Overall these results in the fact that both limit solutions (for each agent) do not give a ground truth prediction for $\boldsymbol{X}_1$ and $\boldsymbol{X}_2$ individually, which one can see from equation (20) with $c = 0$. That is, we need combine the predictions of both agents in a specific way to get ground truth labels for datasets $\mathcal{D}_1$ and $\mathcal{D}_2$. Moreover, the way we combine the solutions differs between datasets $\mathcal{D}_1$ and $\mathcal{D}_2$ that one can see from comparison of right-hand sides of expressions (20) and (21). Given all the above, to predict optimal labels we need to change we way we combine models of agents in dependence on a dataset, but an usual desire is to have one model that predicts ground truth labels for at least both training datasets.

To obtain the expressions for the case of identical models one should 'remove all tildas' in the above expressions and by setting $\boldsymbol{V}_1 = \tilde{\boldsymbol{V}}_1$, $\boldsymbol{V}_2 = \tilde{\boldsymbol{V}}_2$, $\boldsymbol{C}_1 = \tilde{\boldsymbol{C}}_2 \to \boldsymbol{I}$ and $(\boldsymbol{C}_1\tilde{\boldsymbol{P}}_2^\top\tilde{\boldsymbol{C}}_2\boldsymbol{P}_1^\top)^t \to (\boldsymbol{P}_2^\top\boldsymbol{P}_1^\top)^t$

---

[2] In case of $c = 0$, the whole analysis can be repeated by replacing inverse sign with † sign and using the following fact for positive semidefinite matrices (Zhang, 2006):

$$\boldsymbol{V}_2^\top\boldsymbol{D}\boldsymbol{V}_1(\boldsymbol{V}_1^\top\boldsymbol{D}\boldsymbol{V}_1)^\dagger(\boldsymbol{V}_1^\top\boldsymbol{D}\boldsymbol{V}_1) = \boldsymbol{V}_2^\top\boldsymbol{D}\boldsymbol{V}_1.$$

# E  PARALLEL KD

In this section, we theoretically analyze a slight modification of the AvgKD algorithm which we call Parallel KD (PKD). Keeping the notation of sections B and C, denote the data on agent 1 as $\mathcal{D}_1 = (\boldsymbol{X}^1, \boldsymbol{y}^1)$ where $\boldsymbol{X}^1[i,:] = \boldsymbol{x}_n^1$ and $\boldsymbol{y}^1[i] = y_i^1$. Correspondingly, for agent 2 we have $\mathcal{D}^2 = (\boldsymbol{X}^2, \boldsymbol{y}^2)$. Now starting from $\hat{\boldsymbol{y}}_0^1 = \boldsymbol{y}^1, \hat{\boldsymbol{y}}_0^2 = \boldsymbol{y}^2$, in each round $t \geq 0$:

    a. Agents 1 and 2 train their model on datasets $(\boldsymbol{X}^1, \hat{\boldsymbol{y}}_t^1)$ and $(\boldsymbol{X}^2, \hat{\boldsymbol{y}}_t^2)$ to obtain $g_t^1$ and $g_t^2$.
    b. Agents exchange $g_t^1$ and $g_t^2$ between each other.
    c. Agents use exchanged models to predict labels $\hat{\boldsymbol{y}}_{t+1}^1 = \frac{\hat{\boldsymbol{y}}_t^1 + g_t^2(\boldsymbol{X}^1)}{2}, \hat{\boldsymbol{y}}_{t+1}^2 = \frac{\hat{\boldsymbol{y}}_t^2 + g_t^1(\boldsymbol{X}^2)}{2}$.

The summary of the algorithm is depicted in Figure 7b. That is, we learn from the average of 2 agents' predictions. For simplicity, we are going to analyze the scheme without regularization and we take always a minimum norm solution. Notice that there is no exchange of raw data but only of the trained models.

Let us analyse KD algorithm where solutions for agent I ($g^1$) and agent II ($g^2$) obtained as:

$$g_t^1(\mathcal{X}_2) = \frac{1}{2}\boldsymbol{K}_{21}(\boldsymbol{K}_{11})^\dagger(g_{t-1}^1(\mathcal{X}_1) + g_{t-1}^2(\mathcal{X}_1)) = \frac{1}{2}\boldsymbol{V}_2^\top \hat{\boldsymbol{P}}_1^\top(g_{t-1}^1 + g_{t-1}^2), \tag{22}$$

$$g_t^2(\mathcal{X}_1) = \frac{1}{2}\boldsymbol{K}_{12}(\boldsymbol{K}_{22})^\dagger(g_{t-1}^1(\mathcal{X}_2) + g_{t-1}^2(\mathcal{X}_2)) = \frac{1}{2}\boldsymbol{V}_1^\top \hat{\boldsymbol{P}}_2^\top(g_{t-1}^1 + g_{t-1}^2), \tag{23}$$

where

$$g_{t-1}^1 = \frac{1}{2}\boldsymbol{D}\boldsymbol{V}_1(\boldsymbol{K}_{11})^\dagger(g_{t-2}^1(\mathcal{X}_1) + g_{t-2}^2(\mathcal{X}_1)), \quad \forall t \geq 2 \tag{24}$$

$$g_{t-1}^2 = \frac{1}{2}\boldsymbol{D}\boldsymbol{V}_2(\boldsymbol{K}_{22})^\dagger(g_{t-2}^1(\mathcal{X}_2) + g_{t-2}^2(\mathcal{X}_2)), \quad \forall t \geq 2 \tag{25}$$

$$t = 1: \quad g_0^1 = \hat{\boldsymbol{P}}_1^\top \boldsymbol{z}_1 \quad \text{and} \quad g_0^2 = \hat{\boldsymbol{P}}_2^\top \boldsymbol{z}_2, \quad \text{where} \quad \boldsymbol{z}_i = \boldsymbol{V}_i \boldsymbol{y}_i, \quad i = 1, 2. \tag{26}$$

Consider $g_t^1$ for $t \geq 1$:

$$g_t^1 = \frac{1}{2}\hat{\boldsymbol{P}}_1^\top(g_{t-1}^1 + g_{t-1}^2) = \frac{1}{2}\hat{\boldsymbol{P}}_1^\top(\frac{1}{2}\hat{\boldsymbol{P}}_1^\top(g_{t-2}^1 + g_{t-2}^2) + \frac{1}{2}\hat{\boldsymbol{P}}_2^\top(g_{t-2}^1 + g_{t-2}^2)) =$$

$$= \frac{1}{2}\hat{\boldsymbol{P}}_1^\top(\frac{1}{2}(\hat{\boldsymbol{P}}_1^\top + \hat{\boldsymbol{P}}_2^\top)(g_{t-2}^1 + g_{t-2}^2)) = ... = \frac{1}{2^t}\hat{\boldsymbol{P}}_1^\top(\hat{\boldsymbol{P}}_1^\top + \hat{\boldsymbol{P}}_2^\top)^{t-1}(g_0^1 + g_0^2) =$$

$$= \frac{1}{2^t}\hat{\boldsymbol{P}}_1^\top(\hat{\boldsymbol{P}}_1^\top + \hat{\boldsymbol{P}}_2^\top)^{t-1}(\hat{\boldsymbol{P}}_1^\top \boldsymbol{z}_1 + \hat{\boldsymbol{P}}_2^\top \boldsymbol{z}_2)$$

The form of the solutions reminds the method of averaged projection (Lewis et al., 2007) with operator $\hat{\boldsymbol{P}}_1 + \hat{\boldsymbol{P}}_2$, which is similar to alternating projection converges to the intersection point of 2 subspaces[3]. That is, similarly to AKD in the case with the regularization we expect the solution to converge to the origin point $\boldsymbol{0}$ in the limit of the distillation steps. As a result, after some point, one expects steady degradation of the predictions of both agents in the PKD scheme.

# F  ENSEMBLED KD

One can consider the following problem:

$$\begin{pmatrix} \boldsymbol{L}_{11} & \boldsymbol{M}_{12} \\ \boldsymbol{L}_{21} & \boldsymbol{M}_{22} \end{pmatrix} = \begin{pmatrix} \boldsymbol{V}_1^\top \boldsymbol{D}\boldsymbol{V}_1 & \tilde{\boldsymbol{V}}_1^\top \boldsymbol{D}\tilde{\boldsymbol{V}}_2 \\ \boldsymbol{V}_2^\top \boldsymbol{D}\boldsymbol{V}_1 & \tilde{\boldsymbol{V}}_2^\top \boldsymbol{D}\tilde{\boldsymbol{V}}_2 \end{pmatrix} \begin{pmatrix} \boldsymbol{\beta}_1 \\ \boldsymbol{\beta}_2 \end{pmatrix} = \begin{pmatrix} \boldsymbol{y}_1 \\ \boldsymbol{y}_2 \end{pmatrix}, \tag{27}$$

with the following identities:

$$\boldsymbol{V}_1^\top \boldsymbol{D}\boldsymbol{V}_1\boldsymbol{\beta}_1 + \tilde{\boldsymbol{V}}_1^\top \boldsymbol{D}\tilde{\boldsymbol{V}}_2\boldsymbol{\beta}_2 = \boldsymbol{y}_1 \quad \text{and} \quad \boldsymbol{V}_2^\top \boldsymbol{D}\boldsymbol{V}_1\boldsymbol{\beta}_1 + \tilde{\boldsymbol{V}}_2^\top \boldsymbol{D}\tilde{\boldsymbol{V}}_2\boldsymbol{\beta}_2 = \boldsymbol{y}_2. \tag{28}$$

---

[3]Actually, there is an explicit relation between alternating and averaged projections (Lewis et al., 2007).

One can find $\boldsymbol{\beta}_1, \boldsymbol{\beta}_2$ and deduce the following prediction by the model associated with the system (27) for $i = 1, 2$:

$$
\begin{aligned}
\boldsymbol{V}_i^\top \boldsymbol{D} \boldsymbol{V}_1 \boldsymbol{\beta}_1 + \tilde{\boldsymbol{V}}_i^\top \boldsymbol{D} \tilde{\boldsymbol{V}}_2 \boldsymbol{\beta}_2 = \\
\boldsymbol{V}_i^\top \boldsymbol{P}_1^\top (\boldsymbol{I} - \boldsymbol{C}_1 \tilde{\boldsymbol{P}}_2^\top \tilde{\boldsymbol{C}}_2 \boldsymbol{P}_1^\top)^\dagger \boldsymbol{z}_1 - \boldsymbol{V}_i^\top \boldsymbol{P}_1^\top \boldsymbol{C}_1 \tilde{\boldsymbol{P}}_2^\top (\boldsymbol{I} - \tilde{\boldsymbol{C}}_2 \boldsymbol{P}_1^\top \boldsymbol{C}_1 \tilde{\boldsymbol{P}}_2^\top)^\dagger \boldsymbol{z}_2 + \\
\tilde{\boldsymbol{V}}_i^\top \tilde{\boldsymbol{P}}_2^\top (\boldsymbol{I} - \tilde{\boldsymbol{C}}_2 \boldsymbol{P}_1^\top \boldsymbol{C}_1 \tilde{\boldsymbol{P}}_2^\top)^\dagger \boldsymbol{z}_2 - \tilde{\boldsymbol{V}}_i^\top \boldsymbol{P}_2^\top \tilde{\boldsymbol{C}}_2 \boldsymbol{P}_1^\top (\boldsymbol{I} - \boldsymbol{C}_1 \tilde{\boldsymbol{P}}_2^\top \tilde{\boldsymbol{C}}_2 \boldsymbol{P}_1^\top)^\dagger \boldsymbol{z}_1 = \\
\boldsymbol{V}_i^\top \boldsymbol{P}_1^\top \sum_{t=0}^{\infty} (\boldsymbol{C}_1 \tilde{\boldsymbol{P}}_2^\top \tilde{\boldsymbol{C}}_2 \boldsymbol{P}_1^\top)^t \boldsymbol{z}_1 - \boldsymbol{V}_i^\top \boldsymbol{P}_1^\top \boldsymbol{C}_1 \tilde{\boldsymbol{P}}_2^\top \sum_{t=0}^{\infty} (\tilde{\boldsymbol{C}}_2 \boldsymbol{P}_1^\top \boldsymbol{C}_1 \tilde{\boldsymbol{P}}_2^\top)^t \boldsymbol{z}_2 + \\
\tilde{\boldsymbol{V}}_i^\top \boldsymbol{P}_2^\top \sum_{t=0}^{\infty} (\tilde{\boldsymbol{C}}_2 \boldsymbol{P}_1^\top \boldsymbol{C}_1 \tilde{\boldsymbol{P}}_2^\top)^t \boldsymbol{z}_2 - \tilde{\boldsymbol{V}}_i^\top \boldsymbol{P}_2^\top \tilde{\boldsymbol{C}}_2 \boldsymbol{P}_1^\top \sum_{t=0}^{\infty} (\boldsymbol{C}_1 \tilde{\boldsymbol{P}}_2^\top \tilde{\boldsymbol{C}}_2 \boldsymbol{P}_1^\top)^t \boldsymbol{z}_1 .
\end{aligned}
\tag{29}
$$

From this one can easily deduce (28) which means that for datasets of both agents this model predicts ground truth labels. To obtain the expressions for the case of identical models one should 'remove all tildas' in all the above expressions and by setting $\boldsymbol{V}_1 = \tilde{\boldsymbol{V}}_1$, $\boldsymbol{V}_2 = \tilde{\boldsymbol{V}}_2$, $\boldsymbol{C}_1 = \tilde{\boldsymbol{C}}_2 \to \boldsymbol{I}$ and $(\boldsymbol{C}_1 \tilde{\boldsymbol{P}}_2^\top \tilde{\boldsymbol{C}}_2 \boldsymbol{P}_1^\top)^t \to (\boldsymbol{P}_2^\top \boldsymbol{P}_1^\top)^t$

The last question is how one can construct the scheme of iterative KD rounds to obtain the above expression for the limit model. From the form of the prediction, we conclude that one should use models obtained in the process of AKD. There are many possible schemes how one can combine these models to obtain the desired result. One of the simplest possibilities is presented in the section 5.2.

## G    M-AGENT SCHEMES

The natural question to ask is how one could extend the discussed schemes to the setting of M agents. In this section, we address this question and explicitly provide the description of each algorithm for the setting of $M$ agents.

**Scalability and privacy.**    Before diving into particular algorithms we investigate some important concerns about our KD based framework. A naive implementation of our methods will require each agent sharing their model with all other agents. This will incur significant communication costs ($M^2$), storage costs (each agent has to store $M$ models), and is not compatible with secure aggregation (Bonawitz et al., 2016) potentially leaking information. One potential approach to alleviating these concerns is to use an server and homomorphic encryption (Graepel et al., 2012). Homomorphic encryption allows agent 1 to compute predictions on agent 2's data $f_1(\boldsymbol{X}^2)$ without learning anything about $\boldsymbol{X}^2$ i.e. there exists a procedure Hom such that given encrypted data $\mathrm{Enc}(\boldsymbol{X}^2)$, we can compute

$$
\mathrm{Hom}(f_1, \mathrm{Enc}(\boldsymbol{X}^2)) = \mathrm{Enc}(f_1(\boldsymbol{X}^2)) .
$$

Given access to such a primitive, we can use a dedicated server (agent 0) to whom all models $f_1, \ldots, f_M$ are sent. Let us define some weighted sum of the predictions as $f_{\boldsymbol{\alpha}}(\boldsymbol{X}) := \sum_{i=1}^{M} \alpha_i f_i(\boldsymbol{X})$. Then, using homomorphic encryption, each agent $i$ can compute $\mathrm{Enc}(f_{\boldsymbol{\alpha}}(\boldsymbol{X}^i)))$ in a private manner without leaking any information to agent 0. This makes the communication cost linear in $M$ instead of quadratic, and also makes it more private and secure. A full fledged investigation of the scalability, privacy, and security of such an approach is left for future work. With this caveat out of the way, we next discuss some concrete algorithms.

**AKD with M agents.**    To extend AKD scheme to a multi-agent setting and corresponding theory we need to start by understanding what the alternating projection algorithm is in the case of $M$ convex sets. Suppose we want to find the intersection point of $M$ affine sets $\mathcal{C}_i$, for $i = 1, ..., M$. In terms of alternating projection algorithm we can write the following extension of it (Halperin, 1962):

$$
\boldsymbol{P}_{\cap_{i=1}^n \mathcal{C}_i}(\boldsymbol{x}) = (\boldsymbol{P}_{\mathcal{C}_M} \boldsymbol{P}_{\mathcal{C}_{M-1}} ... \boldsymbol{P}_{\mathcal{C}_1})^\infty (\boldsymbol{x})
\tag{30}
$$

For our algorithm, it means that agent 1 passes its model to agent 2, then agent 2 passes its model to agent 3 and so until agent M that passes its model to agent 1, and then the cycle repeats. That is, as before, we denote the data on agent 1 as $\mathcal{D}_1 = (\boldsymbol{X}^1, \boldsymbol{y}^1)$ where $\boldsymbol{X}^1[i, :] = \boldsymbol{x}_n^1$ and $\boldsymbol{y}^1[i] = y_i^1.$, for all other agents we have $\mathcal{D}^i = (\boldsymbol{X}^i, \boldsymbol{y}^i)$, for $i = 2, ..., M$. Now starting from $\hat{\boldsymbol{y}}_0^1 = \boldsymbol{y}^1$, in each rounds $t, ..., t + M - 1$, $t \geq 0$:

    a. Agent 1 trains their model on dataset $(\boldsymbol{X}^1, \hat{\boldsymbol{y}}_t^1)$ to obtain $g_t^1$.

    b. for $i = 2, ..., M$:

        b.1 Agent $i$ receives $g_{t+i-2}^1$ and uses it to predict labels $\hat{\boldsymbol{y}}_{t+i-1}^i = g_{t+i-2}^1(\boldsymbol{X}^i)$.

        b.2 Agent $i$ trains their model on dataset $(\boldsymbol{X}^i, \hat{\boldsymbol{y}}_{t+i-1}^i)$ to obtain $g_{t+i-1}^1$.

    c. Agent 1 receives a model $g_{t+M-1}^1$ from agent M and predicts $\hat{\boldsymbol{y}}_{t+M}^1 = g_{t+M-1}^1(\boldsymbol{X}^1)$.

As before, there is no exchange of raw data but only of the trained models. And given all the results deduced before, all the models from some point will start to degenerate. The rate of convergence of such an algorithm is defined similarly to alternating projection in case 2 sets and can be found in Smith et al. (1977).

**PKD with M agents.** PKD scheme can be easily extended to the multi-agent setting analogously to how averaged projection algorithm can be extended to the multi-set setting (Lewis et al., 2007). Suppose we want to find the intersection point of $M$ affine sets $\mathcal{C}_i$, for $i = 1, ..., M$ then the in terms of averaged projection we have the following:

$$\boldsymbol{P}_{\cap_{i=1}^n \mathcal{C}_i}(\boldsymbol{x}) = (\frac{1}{M}\sum_{i=1}^M \boldsymbol{P}_{\mathcal{C}_i})^\infty(\boldsymbol{x}) \tag{31}$$

This expression easily translates into the PKD algorithm for M agents. Denote the data on all agents as $\mathcal{D}^i = (\boldsymbol{X}^i, \boldsymbol{y}^i)$, for $i = 2, ..., M$. Now starting from $\hat{\boldsymbol{y}}_0^i = \boldsymbol{y}^i$, for $i = 2, ..., M$, in each round $t \geq 0$:

    a. for $i = 1, ..., M$:

        a.1 Agent $i$ trains their model on dataset $(\boldsymbol{X}^i, \hat{\boldsymbol{y}}_t^i)$ to obtain $g_t^i$.

    b. for $i = 1, ..., M$:

        b.1 Agent $i$ receives models $g_t^j$, for $j = 1, ..., M, j \neq i$ from all other agents.

        b.2 Agent $i$ use received models to predict $\hat{\boldsymbol{y}}_{t+1}^i = \frac{\hat{\boldsymbol{y}}_t^i + \sum_{j=1, j \neq i}^M g_t^j(\boldsymbol{X}^i)}{M}$.

As in the case of 2 agents, there is no exchange of data between agents, but only models. This scheme requires all to all communication.

**AvgKD with M agents.** Similarly to PKD algorithm we can extend AvgKD algorithm as follows: starting from $\hat{\boldsymbol{y}}_0^i = \boldsymbol{y}^i$, for $i = 2, ..., M$, in each round $t \geq 0$:

    a. for $i = 1, ..., M$:

        a.1 Agent $i$ trains their model on dataset $(\boldsymbol{X}^i, \hat{\boldsymbol{y}}_t^i)$ to obtain $g_t^i$.

    b. for $i = 1, ..., M$:

        b.1 Agent $i$ receives models $g_t^j$, for $j = 1, ..., M, j \neq i$ from all other agents.

        b.2 Agent $i$ use received models to predict $\hat{\boldsymbol{y}}_{t+1}^i = \frac{\boldsymbol{y}^i + \sum_{j=1, j \neq i}^M g_t^j(\boldsymbol{X}^i)}{M}$.

That is, there is no exchange of data between agents, but only models. This scheme as well as PKD requires all to all communication. That means that the scheme can not be scaled, but it is still useful for small numbers of agents (e.g collaboration of companies). The last is motivated by the simplicity of the scheme without any need for hyperparameter tuning, the non-degrading behavior as well as its superiority over the FedAvg scheme at highly heterogeneous data regimes.

**EKD with M agents.** EKD scheme in the case of 2 agents is based on models obtained in the process of 2 simultaneous runs of the AKD algorithm. This means that the extension of EKD to the multi-agent setting, in this case, is straightforward by using the $M$ simultaneous runs of AKD algorithm starting from each agent in the multi-agent setting and summing the obtained models in the process as follows:

$$f_\infty(\boldsymbol{x}) = \sum_{t=0}^\infty (-1)^t (\sum_{i=1}^M g_t^i(\boldsymbol{x})) \tag{32}$$

## H ADDITIONAL EXPERIMENTS

### H.1 MNIST WITH VARYING DATA HETEROGENEITY

In this section, we present the results for MNIST dataset with varying data heterogeneity in the setting of 'different model'. The results one can see in Fig. 8. There is a faster degradation trend for both AKD and PKD schemes if different models for agents are used (Fig. 8) comparing to rhe 'same model' setting (Fig. 6) at all data heterogeneity regimes. The PKD scheme is a slight modification of the AvgKD scheme which is proven to degrade through rounds of distillation. We see the degradation trend for PKD scheme which is suggested by our theory presented in App. E. EKD does not improve with subsequent rounds in the setting of 'different models'. AvgKD scheme outperforms both PKD and AKD in all settings. However, its convergence is not stable in extremely high heterogeneous settings, showing large oscillations. Investigating and mitigating this could be interesting future work.

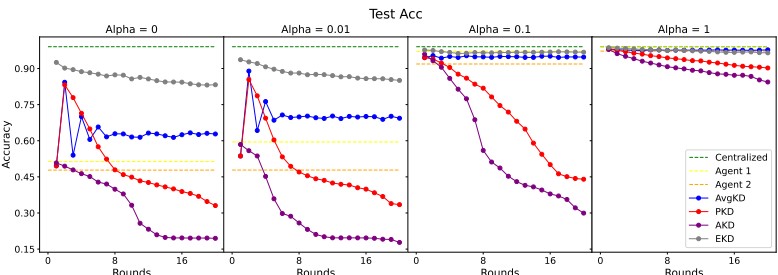

Figure 8: Test accuracy of on MNIST with varying data heterogeneity in the setting of 'different model'. Performance of PKD and AKD degrade with degradation speeding up with the increase in data heterogeneity. Performance of AvgKD scheme converges to steady behavior at any regime of data heterogeneity. Both agents benefit from AvgKD, PKD and EKD schemes in the early rounds of communication.

### H.2 CIFAR10 WITH VARYING DATA HETEROGENEITY

In this section, we present the results for CIFAR10 dataset with varying data heterogeneity. Note that we use cross-entropy loss function here. The results one can see in Fig. 9 and 10 that again show data heterogeneity plays key role in the behavior of all the schemes. All the trends we saw on the MNIST dataset are repeated here except one: EKD does not improve in subsequent rounds in the 'same model' setting.

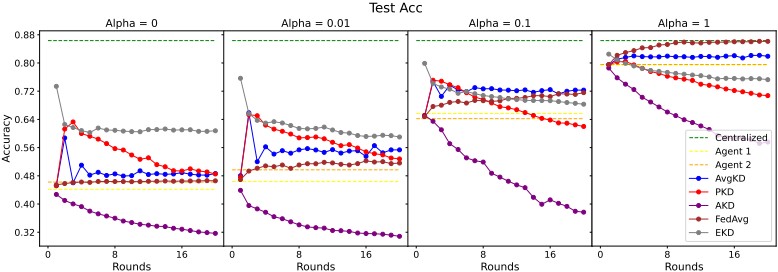

Figure 9: Test accuracy of on CIFAR10 with varying data heterogeneity in the setting of 'same model'. As on MNIST: performance of PKD and AKD degrade with degradation speeding up with the increase in data heterogeneity; performance of AvgKD scheme converges to steady behavior; both agents benefit from AvgKD, PKD and EKD schemes in early rounds of communication.

### H.3 CROSS-ENTROPY OBJECTIVE

In this section, we present the results of experiments on MNIST with Cross-Entropy (CE) loss for 2 main schemes under investigation: AKD and AvgKD. In Fig. 11 and 12 one can see the results of AKD and AvgKD schemes correspondingly for CE loss. The results are aligned with our theory: in

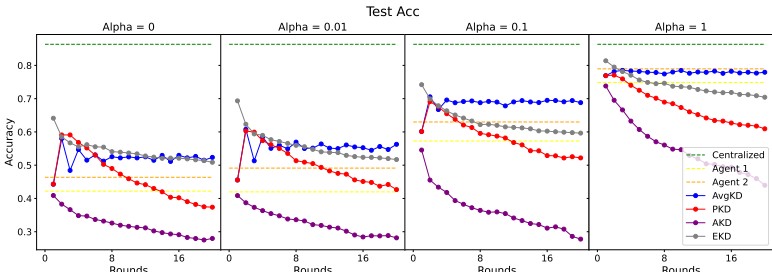

Figure 10: Test accuracy of on CIFAR10 with varying data heterogeneity in the setting of 'different model'. All the schemes behave similarly to the 'same model' setting.

Fig. 11 we see the degradation trend for AKD which is dependent on the amount of the regularization, model and data heterogeneity. In Fig. 12 we see steady behavior of AvgKD scheme for both agents models: there is no degradation even if model and data are different.

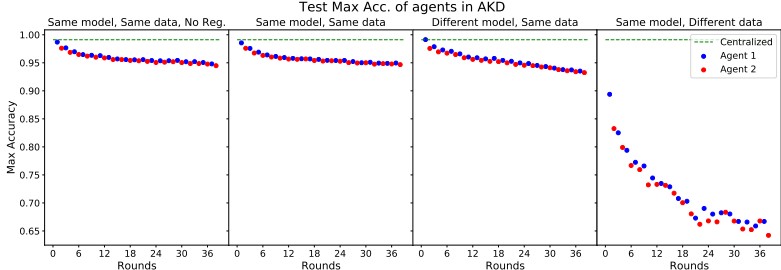

Figure 11: Test accuracy of AKD on MNIST using CE loss and model starting from agent 1 (blue) and agent 2 (red) with varying amount of regularization, model heterogeneity, and data heterogeneity. In all cases, performance degrades with increasing rounds with degradation speeding up with the increase in regularization, model heterogeneity, or data heterogeneity.

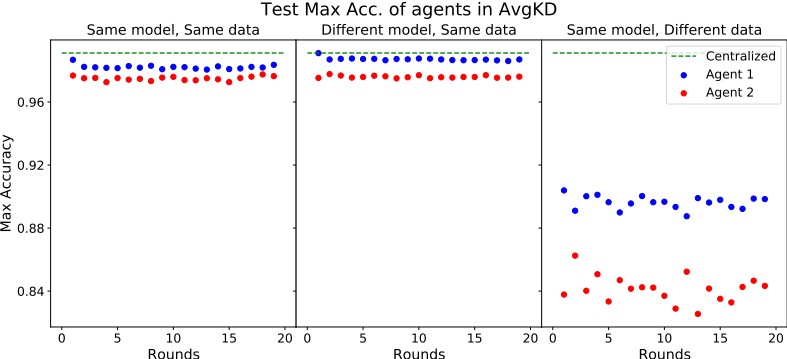

Figure 12: Test accuracy of AvgKD on MNIST using CE loss and model starting from agent 1 (blue) and agent 2 (red) with varying model heterogeneity, and data heterogeneity. In all cases, there is no degradation of performance, though the best accuracy is obtained by agent 1 in round 1 with only local training.

## H.4 COLLABORATION BETWEEN MLPS AND RANDOM FORESTS

In this section, we present the results for MNIST dataset for Random Forests (RF) and MLP models with MSE loss. That is, the experiments are in the setting 'different model', where agent 1 has MLP model and agent 2 has RF model. These experiments can show how AKD and AvgKD schemes behave in the setting of fundamentally different models. In the Figs. 13 and 14 the results for accuracy

are presented. We see the alignment of these results with theory and other experiments with deep learning models. Mainly, there is a degradation trend for AKD scheme which is speeding up with the increase in data heterogeneity, there is no degradation for AvgKD scheme, and the performance of both agents in AvgKD scheme is highly dependent on data heterogeneity.

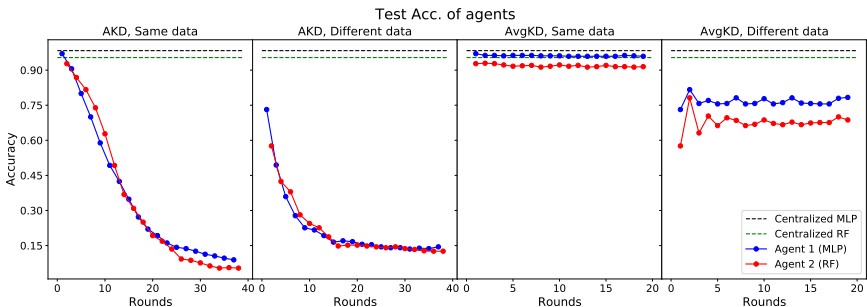

Figure 13: Test accuracy of AKD and AvgKD on MNIST using models MLP (blue) and RF (red) with varying data heterogeneity. For AKD performance degrades with increasing rounds. Degradation is speeding up with the increase in data heterogeneity. For AvgKD there is no degradation of performance.

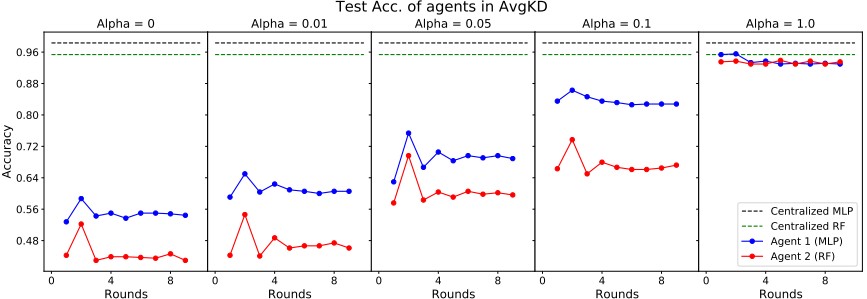

Figure 14: Test accuracy of AvgKD on MNIST using models MLP (blue) and RF (red) with varying data heterogeneity. The increase in data heterogeneity lowers the performance of both agents without degradation trend through rounds.

### H.5 AVGKD WITH M AGENTS

The AvgKD scheme does not degrade in comparison with AKD and PKD schemes that degrade already in the case of 2 agents. In this section, we present the results of the AvgKD scheme with M agents and use 5 agents on the MNIST dataset in the setting of the 'same model' with varying data heterogeneity. In case of full data heterogeneity ($Alpha = 0$) we assign the labels $(2(i-1), 2i-1)$ to the actor number $i$, for $i = 1...5$. The results are presented in the Fig. 15. We see that all the agents repeat the behavior pattern in all cases of data heterogeneity. In cases of $Alpha < 0.05$ (high data heterogeneity) early stopping is beneficial.

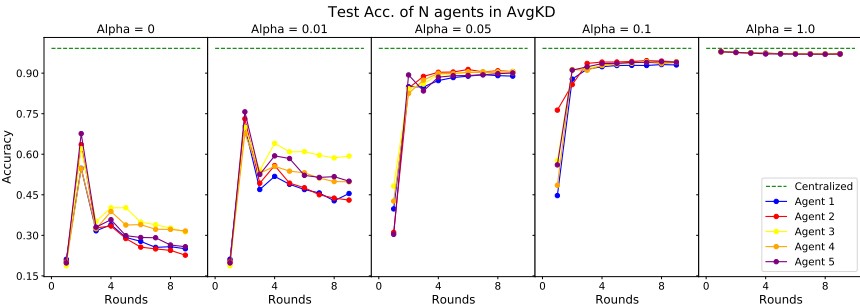

Figure 15: Test accuracy of AvgKD with M agents on MNIST with varying data heterogeneity in the setting of 'same model'. All agents can benefit from the distilled knowledge in early rounds of communication.

