# OpenReview forum: "Towards Model Agnostic Federated Learning Using Knowledge Distillation"
_ICLR.cc/2022/Conference — ICLR 2022 Poster_

### Official Review · Reviewer_1yWX · 2021-11-01

**Correctness:** 3
**Technical Novelty And Significance:** 2
**Empirical Novelty And Significance:** 2
**Recommendation:** 3
**Confidence:** 4

**Main Review:**

NOVELTY & SIGNIFICANCE

This paper uses a simple 2-client federated regression scenario with simplified modeling choices (i.e., kernelized linear regression) with closed-form solutions to develop novel insights which can potentially be translated into new model-agnostic FL framework with strong asymptotic guarantees. The idea (as I summarized above) is novel to me. However, in terms of its practical significance, I am not convinced because of there is too much gap between the theoretic results and real-world setting, which is elaborated below.

1. The theoretic setup started with 2-client setting but has not been extended to multi-agent setting. This is problematic because the result derived in 2-client setting is very specific to the iteration between the two clients and how they average between the local data & the distilled prediction of the shared models. If there are more than one shared models, how would the client set up their training output for the next iteration? Furthermore, the authors assume implicitly that all clients have the same amount of data, which is clearly not true in practical setting -- how will it impact the averaging scheme in such cases?

2. Another peculiar restriction is that the presented result is tied to an obvious flaw in the way AKD is set up in this paper. From the 4 steps mentioned at the beginning of Section 4, client 2 will never get to see true output. I suspect this is a main cause that leads to AKD's degeneration. Given this, the theoretic result developed in 4.2 is kind of moot. Can the authors re-visit this result in case models at iteration t + 1 of any clients are built on distillation of models from iteration t instead -- this is so client 2 can see its true training output at iteration 1.

3. The authors also proposed a data re-incorporation scheme (i.e., AvgKD) which performs better than AKD but given the 2nd point above, it is not clear if this mechanism is necessary if the peculiar flaw in AKD is fixed. My point is if this new mechanism ends up fixing only the flaw above and not anything else, then it is somewhat unnecessary because the above flaw can be fixed by allowing both client to share models concurrently rather than iteratively. Could the authors elaborate more on this?

4. The fix in 3. is later abandoned and replaced with a better EKD fix via ensembling intermediate AKD models. But again, this result is correct given the flawed setup of AKD as pointed out in 2. above. If this flaw is corrected, can the authors revisit the asymptotic convergence result for EKD?

5. The result is largely based on a formulation of regression model and the result is certainly tied to this specific regression form. I am not sure if it is reasonable to anyhow impose the implication of the derived result on a very distant classification setting. In the same vein of thought, another minor restriction is that if we view this regression formulation from a probabilistic perspective then it appears the authors impose the same Gaussian likelihood across all clients and that kind of clashes with the model-agnostic motivation.

6. Have the authors considered the communication cost beyond the 2-client setting? As the distillation requires access to local data, every client would have to send models to every other clients. The total communication cost is therefore N times for than the normal cost of FL and in addition, there will also be extra distillation expense, which is a lot more costly than aggregating model weights.

SOUNDNESS

The results appear correct to me. But I have reservation about the theoretic implication of both AvgKD and EKD as I pointed out above: I am not sure if these proposals were meant to fix anything more than the obvious flaw in the presented AKD setup (e.g. client 2 never gets to see true training output). Furthermore, if the AKD setup were to be fixed by letting client 2 sees its true training output, would the convergence result of EKD still hold?

CLARITY

The paper is well-organized with a nice narrative flow that is easy to follow. But on a minor note, there are a number of grammatical errors all over the paper.

EXPERIMENT

I find the experiment somewhat strange. It implies the initial local model of client 1 is already on the same level with the centralized model which means distillation does not help at all. In all 3 experiment settings, not a single one shows that distillation is being helpful. Am I missing something here?

On another note, the flaw setup on AKD is observable from the reported results. In all 3 settings, client 2 always perform worse than client 1. In fact, on the same model, same data setting, it is noticeable that client 2 is much worse than client 1 at the initial round. This is clearly because client 2 never sees its true training output and this seems to be the case that with the flaw, client 2 is initiating and leading the distillation degradation.


**Summary Of The Paper:**

This paper introduce an interesting perspective on federated learning via knowlegde distillation, which allows participating clients to have their own choices of models. In particular, the paper develops theoretical results for a two-client federated regression scenario, which demonstrates (a) the degeneration of an alternating knowledge distillation where iterative distillations (i.e., a client model at each iteration is re-trained based on unlabeled input + prediction of the other client's model of the previous iteration) gradually lose information over the iterations and eventually converge towards a vacuous model; and (b) a new ensembling technique that aggregates intermediate models produced by both clients over the iterations such that in the limit (i.e., when the no. of iterations tends to infinity), the aggregated model is the same as the (oracle) centralized model. This developed intuition on ensembling intermediate models is then applied to more realistic 2-client federated classification scenarios on MNIST and CIFAR-10 datasets.

**Summary Of The Review:**

This paper presents interesting thoughts that were substantiated via a theoretical exercise on a simplified setting. But unfortunately, in both theory & practice, this is still pretty much a work in progress. Given the huge gap between the theoretic and real-world setup, it is not clear whether one can apply the insight derived from the theoretic setting to another distant real-world setting -- please see my points in 1-5 above. Thus, while this could be the beginning of an interesting theory, more work is needed to complete the idea. As a matter of fact, the current experiment implies the proposed distillation is not working. In addition, I believe the authors have not considered the compute/communication cost of this distillation scheme.

---

> ### Author Response · Authors · 2021-11-22
> **Response (1st part) to Reviewer 1yWX**
>
> We thank the reviewer for the valuable feedback and address the comments as follows:
>
> - 1.a The theoretic setup started with 2-client setting but has not been extended to multi-agent setting. This is problematic because the result derived in 2-client setting is very specific to the iteration between the two clients and how they average between the local data & the distilled prediction of the shared models. If there are more than one shared models, how would the client set up their training output for the next iteration?
>
> We added section G where we extend all the algorithms to the setting with multiple clients. The proofs can similarly be extend quite easily to this new setting, but the computation does not yield any new insight - most of the difficulty is already captured by 2 clients. Thus, we chose to present our framework in this simpler setting. We also show experimental evaluation of AvgKD with 5 clients in Appendix H.6.
>
> - 1.b Furthermore, the authors assume implicitly that all clients have the same amount of data, which is clearly not true in practical setting -- how will it impact the averaging scheme in such cases?
>
> Thank you for pointing out this typo. Our theory does not need any assumption on having equal data. This can also be seen from the derivations in the Appendix. In fact, our experiments utilized a 70-30 split of the data between the two agents (with agent 1 getting 70%). We did this to exactly see if there is an effect on the performance. You may notice that agent 1 tends to have better performance than 2 in some settings because of this, but otherwise our algorithms remain unaffected.
>
> - 2 and 3. Can we fix the data forgetting flaw in AKD by using both models in parallel instead of alternating between them?
>
> This is a great observation. We had initially analyzed this scheme as well (which we called Parallel KD). Theoretically, we saw that it also had degradation and converged to 0. So we removed this for simplicity. We added this back in Appendix E. Our theory also explains why PKD does not solve the degradation issue. This is because, if AKD corresponds to alternating projection $(\Pi_1 \Pi_2)^t$ to find intersection of convex sets, PKD corresponds to averaged using projection $(\frac{\Pi_1 + \Pi_2}{2})^t$. However, this too converges to the intersection (which is 0) and is in fact very similar to alternating projection (see [Sec. 3 of Boyd’s lecture](https://web.stanford.edu/class/ee392o/alt_proj.pdf)). Knowledge distillation inherently tends to “forget” information. This is because it corresponds to a “projection” of the knowledge in model 1 on the data 2, and some information is lost in this process.  So the only way to make sure we do not degrade to 0 is to repeatedly “re-inject” labels as AvgKD does.
>
> The degradation of PKD is also confirmed experimentally in H.7. Though it does initially perform better than AKD, AvgKD outperforms both AKD and PKD.
>
> - 4. The fix in 3. is later abandoned and replaced with a better EKD fix via ensembling intermediate AKD models. But again, this result is correct given the flawed setup of AKD as pointed out in 2. above. If this flaw is corrected, can the authors revisit the asymptotic convergence result for EKD?
>
> The fix of AvgKD was not abandoned, but instead is shown to be limited. AvgKD does not degrade to 0 like AKD or PKD do, but it does not converge to the ideal centralized solution either. As seen from Fig. 6, AvgKD does improve performance over simply training alone, but does not match the centralized target accuracy. In fact, the gap between the centralized target and AvGKD increases with increasing data heterogeneity (smaller alpha).
>
> Given this, we investigate is it at all possible to match centralized accuracy? Towards this we propose EKD which is an *ensembling* algorithm. EKD does theoretically improve performance but requires a large ensemble (upto 500 models). This is not really practical, and does not solve the original goal we set for ourselves - we want to train a *single* model which matches centralized accuracy. EKD is more of a test of our theoretical framework than a useful algorithm in itself.
>
> In conclusion, there does not seem to be any algorithm (AKD, PKD, or AvgKD) which trains a single model matching centralized performance when there is extreme data heterogeneity. This is a fundamental limitation of the knowledge distillation step where information is lost. However, for practical applications with limited heterogeneity, AvgKD seems to provide reasonable performance. It also significantly outperforms FedAvg even when all agents use identical models (see Fig. H.7)

---

> > ### Author Response · Authors · 2021-11-22
> > **Response (2nd part) to Reviewer 1yWX**
> >
> > - 5. The result is largely based on a formulation of regression model and the result is certainly tied to this specific regression form. I am not sure if it is reasonable to anyhow impose the implication of the derived result on a very distant classification setting. In the same vein of thought, another minor restriction is that if we view this regression formulation from a probabilistic perspective then it appears the authors impose the same Gaussian likelihood across all clients and that kind of clashes with the model-agnostic motivation.
> >
> > We agree that our theoretical formulation is simplistic - but this was by choice. Our goal was to develop a framework which is tractable, yet helps us understand and design new algorithms. We believe we achieve this - the theoretical predictions by our framework about AKD, PKD, AvgKD, and EKD closely match the empirical performance on much more complicated deep learning settings. Analyzing the latter directly seems out of reach of current theory (unless other strongly simplifying assumptions are made). Our framework fully allows each agent to use whatever model (and kernel) they wish. So there is no common prior across agents.
> >
> > - 6. Have the authors considered the communication cost beyond the 2-client setting? As the distillation requires access to local data, every client would have to send models to every other clients. The total communication cost is therefore N times for than the normal cost of FL and in addition, there will also be extra distillation expense, which is a lot more costly than aggregating model weights.
> >
> > This is an excellent observation - a naive implementation of our algorithms would require every agent sending their model to every other agent. This would require $O(M)$ memory on each agent and $O(M^2)$ total communication, making it unscalable. However, this can be circumvented by using a trusted server - all $M$ agents can send their models to the server. The agents can send their data and receive predictions back. This approach is way more scalable and matches the costs of FedAvg. However, it is not private since it requires the agents sending their data to the server to give predictions. This loss of privacy can be overcome using cryptographic tools like homomorphic encryption, ensuring both scalability and privacy. We added a discussion on this in Appendix G.
> >
> >
> > **Experimental performance.**
> >
> > - I find the experiment somewhat strange. It implies the initial local model of client 1 is already on the same level with the centralized model which means distillation does not help at all. In all 3 experiment settings, not a single one shows that distillation is being helpful. Am I missing something here?
> >
> > Our initial experiments were overly simplistic and considered unrealistic data distributions. When data is completely iid, or completely non-iid, KD does not have very good performance. However, as seen in Fig. 6, at intermediate levels of data heterogeneity AvgKD does in fact improve the performance over local-only training. AKD on the other hand always shows degrading performance (Fig. 8).
> >
> >
> > - In all 3 settings, client 2 always perform worse than client 1. In fact, on the same model, same data setting, it is noticeable that client 2 is much worse than client 1 at the initial round. This is clearly because client 2 never sees its true training output and this seems to be the case that with the flaw, client 2 is initiating and leading the distillation degradation.
> >
> > In this setting we did a 70-30 split of the data with client 2 having only 30%. This was done to investigate the effect of imbalanced data distribution. The imbalance is the main reason for the differences observed. In Figure 8 (Alpha = 1) we report the results of the iid and equal data split (50-50) between 2 agents. In the initial rounds, there is not much difference between the 2 agents while we still observe the degradation trend through rounds that is suggested by our theory. We apologize for not highlighting how the data was split among the agents - we will rectify this and make this clearer and redo all experiments with a more standardized splitting.

---

> > > ### Author Response · Authors · 2021-11-22
> > > **Summary of response**
> > >
> > > The major concerns raised were i) performance of PKD vs. AvgKD, ii) mismatch between theory and practice.
> > > The first is fully answered by our discussion of PKD. For the latter, we want to emphasize that in all our experiments, the empirical observations closely match our theoretical predictions. This is despite the fact that our theory is on an arguably simplistic setup. Thus, our formulation seems to capture all the essential complexity of the real world performance of the algorithms while still being tractable. We hope the reviewer is sufficiently convinced and if so request them to re-evaluate our work.

---

### Official Review · Reviewer_mYLp · 2021-11-02

**Correctness:** 4
**Technical Novelty And Significance:** 3
**Empirical Novelty And Significance:** 3
**Recommendation:** 8
**Confidence:** 4

**Main Review:**

The paper thoroughly analyzes the dynamics of co-distillation and the proposed variants in a limited setting with two clients. The local models may differ in their kernel function so that model averaging is not possible. While the constrained setup limits practical insights, it allows to clearly analyze the behavior both theoretically and empirically. To that end, the paper details the dynamics of all three approaches and shows that the straight-forward approach may degenerate, and that the ensemble approach is optimal in the limit. Moreover, it gives conditions under which the averaging approach degenerates. These theoretical results are interesting and insightful. The empirical evaluation confirms the theoretical findings and in addition shows that the behavior is similar when using neural networks instead of kernel regression. This hints at the generality of this analysis. The paper is well-written and clear, the theoretical results are sound and, to the best of my knowledge, correct.

I really appreciate the theoretical contributions, but I feel that at least the empirical evaluation is too limited. While there is good reason to analyze only two clients theoretically, it would be straight-forward to apply the proposed methods on larger numbers of clients empirically. Using only two clients and only linear regression and MNIST as experiments limits the contribution. The paper would benefit greatly from a broader empirical evaluation.

Detailed comments:
- Why is only linear regression used for the kernel experiments?
- Which kernels are used for the experiments? I guess, a linear kernel would be most suitable to the task, but then both kernel functions would be the same.
- How does co-distillation compare to averaging kernel models [1]?
- The approach in [2] similarly uses co-distillation. Even though they use an unlabeled reference dataset, it seems worthwhile to compare to it.

[1] Kamp, Michael, et al. "Communication-efficient distributed online learning with kernels." Joint European Conference on Machine Learning and Knowledge Discovery in Databases. Springer, Cham, 2016.
[2] Bistritz, Ilai, Ariana Mann, and Nicholas Bambos. "Distributed distillation for on-device learning." Advances in Neural Information Processing Systems 33 (2020).


******** after rebuttal **********

The authors have addressed my questions to my satisfaction. While I agree with my fellow reviewers that this work is limited in its scope, I do enjoy this novel theoretical take on distributed learning with different model types via knowledge distillation. Thus, I vote for acceptance. I have updated my score accordingly.

**Summary Of The Paper:**

The paper analyzes the dynamics of optimizing two kernel regression models via co-distillation in a distributed setup where local models may differ in the kernel used. In each round, each local client uses the other client's model to produce novel labels for its local dataset and then retrains its local model using these novel labels. The paper analyzes three variants of this approach, the vanilla variant, a variant where novel labels are an average of the actual label and the one predicted by the other client's model, and an ensembling approach that uses all model iterations to produce predictions. The approaches are analyzed theoretically and empirically.

**Summary Of The Review:**

The paper presents an interesting theoretical analysis of co-distillation for two different kernel regression models. The empirical evaluation confirms the theoretical findings. The constrained setting on the one hand allows for strong theoretical results, but on the other limits the practical insights that can be drawn from them. Here, a broader empirical analysis would have improved the significance of the contributions.

---

> ### Author Response · Authors · 2021-11-22
> **Response to Reviewer mYLp**
>
> We thank the reviewer for the valuable feedback and address the comments as follows:
>
> - Empirical evaluation is too limited. While there is good reason to analyze only two clients theoretically, it would be straight-forward to apply the proposed methods on larger numbers of clients empirically. Using only two clients and only linear regression and MNIST as experiments limits the contribution.
>
> Appendix H shows results for cross-entropy loss as well as the results on CIFAR10. We also added results for AvgKD algorithm with 5 agents (H.6), and also show that our framework works even when using completely different models such as Random Forests and Multi-Layer Perceptron (H.5). We also performed more fine-grained investigation into the effect of data heterogeneity on all these algorithms. Finally, we added a comparison of AvgKD scheme with FedAvg. All of these experiments are consistent with our original claims: AvgKD handles model heterogeneity quite well and data heterogeneity is the key bottleneck to using KD in model agnostic federated learning.
>
> - Why is only linear regression used for the kernel experiments?
>
> For the toy experiments we indeed used a linear kernel. But for experiments on MNIST and CIFAR10 datasets we used Convolutional Neural Networks, Multi-Layer Perceptron and Random Forests.
>
> - How does co-distillation compare to averaging kernel models [1]?
>
> Thank you for the very interesting reference. Our setup differs from Kamp et al. in two important aspects: firstly they require that all the different distributed workers use the same kernel whereas we do not want to impose such a restriction. But perhaps more importantly, their algorithms of compressing and communicating the kernels is highly specific to kernels. We do not see an obvious way to use this approach for deep learning training. Note that our kernel framework is used only to provide insights. We want our algorithms to be applicable to practical deep learning settings.
>
> - The approach in [2] similarly uses co-distillation. Even though they use an unlabeled reference dataset, it seems worthwhile to compare to it.
>
> Thank you for the reference - we will be sure to add it to our related work discussion. As noted, codistillation in its original form is not applicable to our setting since there is usually no shared reference data available in federated learning. Adapting codistillation to our setting would yield a algorithm similar to but not identical to AvgKD. We call it parallel KD and added a theoretical analysis in Appendix E, and experimental comparison in Appendix H.7. We see that PKD is better than AKD, but worse than AvgKD. Just like AKD, it eventually degrades over time converging to 0.

---

### Official Review · Reviewer_fTfB · 2021-11-03

**Correctness:** 4
**Technical Novelty And Significance:** 4
**Empirical Novelty And Significance:** 4
**Recommendation:** 6
**Confidence:** 3

**Main Review:**

Pros:
1. This work gives the first theory analysis for model agnostic FL, even though the setting is very simple. They also use the negative result to motivate the improved algorithm ensemble AvgKD. The whole story is complete and clear.

2. The experiments are sufficient, and support their analysis.


Cons:
1. Lin et al also have similar KD idea for model agnostic FL. What is difference between AvgKD and their work?
Lin, Tao, et al. "Ensemble distillation for robust model fusion in federated learning." arXiv preprint arXiv:2006.07242 (2020).

2. As I mentioned before, the setting is somewhat toy. In practice, there are multiple agents, and the machine learning model is way complicated, that may not have closed form solution.

3. In addition, the paper only consider optimization perspective, but I am curious about the generalization ability of AvgKD. Does it provably generalize better than purely local training?

**Summary Of The Paper:**

This paper analyze knowledge distillation based model agnostic federated learning. They consider simple two agent kernel regression scenario, where each agent has its own dataset and predicting function. They propose to train agent 1 model on dataset 1, and then use agent 1's model to make predictions on dataset 2, and agent 2 will train model using these predictions. They analyze the dynamics of agent 1's model, and show that Alternating Knowledge distillation will degrade the model prediction to 0. They provide another algorithm ensemble AvgKD which can actually avoid this issue and converge to optimal solution. The experiments also show that AvgKD can converge to zero loss while other approaches do not work.

**Summary Of The Review:**

Overall, model agnostic FL area is lack of theory results, so this work has its own novelty. However, the setting is somewhat toy, so I am afraid it may not be very helpful towards understanding model agnostic FL in practice (e.g., multiple clients and complicated models that do not have closed form solution.)

---

> ### Author Response · Authors · 2021-11-22
> **Response to Reviewer fTfB**
>
> We thank the reviewer for the valuable feedback and address the comments as follows:
>
> - Comparison with Lin, Tao, et al. "Ensemble distillation for robust model fusion in federated learning." arXiv preprint arXiv:2006.07242 (2020).
>
> The mentioned paper utilizes public data which is not the case for our methods. In our methods agents only transfer models to each other and train their model using their local data. Moreover, the algorithm proposed in this paper in the simple 2 agent setting with private X becomes AKD, which as we show, does not perform well.
>
> - In practice,there are multiple agents, and the machine learning model is complicated and may not have a closed form solution.
>
> Section on multi-agent settings for each analyzed in our paper scheme is added to the Appendix along with corresponding extensions of theory. The experiments for AvgKD scheme with 5 agents are added too. Moreover, the experiments with completely different models as Random Forests and Multi-Layer Perceptron are added to the Appendix. Having said this, indeed our theoretical analysis setting is simplistic. But this is by choice to make it possible to derive meaningful insights. Our paper shows that even this simplistic setting is very useful and its predictions closely match experimental performance on real world models and datasets.
>
> - Does AvgKD provably generalize better than purely local training?
>
> All the results of experiments we present are for the test data, but indeed our theory is only for training. Investigating generalization of AvgKD is very interesting future work.

---

> > ### Comment · Reviewer_fTfB · 2021-12-02
> > **Thanks for providing comments and new results**
> >
> > I appreciate authors' efforts on addressing my concerns. My main concern is still that the setting is bit toy. I keep my score, weakly accept.

---

### Official Review · Reviewer_w5jn · 2021-11-06

**Correctness:** 3
**Technical Novelty And Significance:** 3
**Empirical Novelty And Significance:** 2
**Recommendation:** 6
**Confidence:** 3

**Main Review:**

Strengths:
* This paper introduced the federated kernel regression framework where they formalized notions of both model and data heterogeneity, which can be useful for developing new algorithms for model agnostic federated learning.
* The theoretical analysis for knowledge distillation, including AKD, AvgKD, and EKD is very rigorous.
Weakness:
* The application of the proposed method is limited. Based on the problem setting, the proposed method is only applicable for the case of two agents, while the problem of federated learning is usually for multi-agents (more than two). Can this proposed method be extended to a more agents case?
* The analysis of the experimental section is not clear enough or even missing:
Ex1: For Figure 3, is the loss of AKD convergent or not? This result cannot be directly observed from the figure. This paper can give an analysis on this and also for the reason why the loss of AKD increase.
Ex2: The analysis for Figure 4, 5, 6 is missing, which can make the readers confused about these results.
* Minor problem:
Some of the notations in the paper are a bit confusing to me, especially those seem to have similar meanings. I would suggest that if have the same meaning, they can be represented by the same notation， if not, more explanations for their differences can be offered.
Ex: It seems that $h(t)$ in Algorithm AKD have similar meaning as $g_t^2$ in Algorithm AvgKD (because these two algorithms have similar schemes).

**Summary Of The Paper:**

This paper focuses on the setting of federated learning where the two agents are attempting to perform kernel regression using different kernels (and hence have different models). Their study yields an algorithm of using alternating knowledge distillation (AKD) imposes overly strong regularization and may lead to severe under-fitting. Their theory also shows an interesting connection between AKD and the alternating projection algorithm for finding the intersection of sets. Leveraging this connection, they propose an algorithm that improves upon AKD.

**Summary Of The Review:**

* Although this paper introduced the federated kernel regression framework which might be useful for developing new algorithms for model agnostic federated learning, however, the application of the proposed method is limited.
* Although the theoretical analysis part is rigorous, the analysis of the experimental section is not clear enough or even missing.

---

> ### Author Response · Authors · 2021-11-22
> **Response to Reviewer w5jn**
>
> We thank the reviewer for the valuable feedback and address the comments as follows:
> - Can this proposed method be extended to a more agents case?
>
> We added the section to the Appendix which addresses this question for every scheme we analyzed in this paper. As well as the experiments for AvgKD scheme with 5 agents.
>
> - For Figure 3, is the loss of AKD convergent or not?
>
> Figure 3 is supposed to show the main trends which we derive from the theory: AKD degenerates, AvgKD stabilizes after some round, EKD achieves Centralized model performance.
>
> - The analysis for Figure 4, 5, 6 is missing, which can make the readers confused about these results.
>
> The discussion of the results presented in Figures 4,5,6 is added. Overall the experiment section is extended and more explanations are added.
>
> - h(t) in Algorithm AKD have similar meaning as gt2 in Algorithm AvgKD
>
> Corrected.

---

### Author Response · Authors · 2021-11-22
**Updates made to the paper and some general comments**

We thank all the reviewers for their valuable feedback. Based on this, we have expanded our  experimental evaluation:

1. **Parallel KD** (PKD): we analyze another variant PKD in Appendix E, and see its empirical performance in Fig. 17. This can be seen as a variant “in between” AKD and AvgKD. However, exactly as predicted by theory, PKD still suffers from degradation. This yet again validates our theoretical framework with the experiments and theory matching each other.

2. **Fine grained data heterogeneity**: we introduce a parameter Alpha which controls the amount of non-iidness (alpha = 0 is completely separated by labels and alpha=1 is iid). In this setup, we evaluate AvgKD in Fig. 6, AKD in Fig. 8, and PKD in Fig. 17. We observe that AKD shows degradation consistently across all settings. AvgKD actually performs quite well for reasonable heterogeneity with very fast convergence. However, the gap between AvgKD and the centralized target increases with data heterogeneity.

3. **Comparison with FedAvg**: when agents use the same model, we also run and compare with FedAvg. As we see in Fig. 17, FedAvg is always slower than AvgKD showing that the latter is perhaps a better approach to federated learning in general - not just when each user has different models.

4. **Collaboration between MLP and RF**: we showcase the flexibility of the model agnostic framework by training an MLP on agent 1 and a random forest on 2 (Fig. 14 and 15). Even in this setup, our theoretical predictions hold up - AKD degrades, AvgKD does not, performance gets worse with increasing data heterogeneity. AvgKD does slightly improve over local only training, but new algorithms are needed to match centralized accuracy.

5. **Extension to M agents**: we show how to extend our algorithms to more than 2 agents in Appendix G and evaluate them in H.6. The extensions are straightforward and the results are qualitatively the same as the 2 agent case.

The major concerns raised by the reviewers were i) performance of PKD vs. AvgKD, ii) more thorough experimental evaluation, and iii) scalability to M agents. With this update, we have addressed all of these concerns. There is definitely more room for improvement, and we would be glad to incorporate any other suggestions the reviewers have to offer in the future versions. We want to emphasize that in all our experiments, the *empirical observations closely match our theoretical predictions*. This is despite the fact that our theory is on an arguably simplistic setup. Thus, our formulation seems to capture all the essential complexity of the real world performance of the algorithms while still being tractable.

---

### Decision · Program_Chairs · 2022-01-20

**Decision:**

Accept (Poster)

**Comment:**

This manuscript proposes and analyzes a distillation approach to address heterogeneity in distributed learning. The main paper focuses on a relatively simple two-agent kernel regression setting, and the insights developed are extended (and partially analyzed) for a multiagent setting.

There are four reviewers, all of whom agree that the method addresses an interesting and timely issue. However, reviewers are mixed on the paper score. While all reviewers agree that the setting is somewhat stylized, a subset of reviewers highlights that the results give some deep insight that might drive future analysis and implementation in the area. Other concerns raised include potential issues with the communication overhead and the simplicity of the kernel regression setting vs real-world deep learning. There are initial concerns about whether the failure case is realistic, which the authors address. Extensions to the multi-agent setting and a partial analysis are also addressed by the authors and partially satisfy the reviewers. Nevertheless, after reviews and discussion, the reviewers are mixed at the end of the discussion.

The area chair finds, first, that the paper is much improved, and much more applicable in the updated form than in the original version, and indeed, the insights from the simple model may be informative for practice. However, the concerns raised about the distance between theory and practice are valid. The final opinion remains borderline. Authors are encouraged to address the highlighted technical concerns in any future submission of this work. In particular, the muti0agent setting should probably be central in the discussion of this work. More ambitious empirical evaluation showing that the theory translates to practice )even if there is a gap) would also help.